# Bismuth antimicrobial drugs serve as broad-spectrum metallo-β-lactamase inhibitors

Runming Wang [1,2], Tsz-Pui Lai[1], Peng Gao[2,3], Hongmin Zhang [1,4], Pak-Leung Ho[2,3,5], Patrick Chiu-Yat Woo[2,3,5], Guixing Ma[4], Richard Yi-Tsun Kao[2,3,5], Hongyan Li[1] & Hongzhe Sun [1]

Drug-resistant superbugs pose a huge threat to human health. Infections by Enterobacteriaceae producing metallo-β-lactamases (MBLs), e.g., New Delhi metallo-β-lactamase 1 (NDM-1) are very difficult to treat. Development of effective MBL inhibitors to revive the efficacy of existing antibiotics is highly desirable. However, such inhibitors are not clinically available till now. Here we show that an anti-*Helicobacter pylori* drug, colloidal bismuth subcitrate (CBS), and related Bi(III) compounds irreversibly inhibit different types of MBLs via the mechanism, with one Bi(III) displacing two Zn(II) ions as revealed by X-ray crystallography, leading to the release of Zn(II) cofactors. CBS restores meropenem (MER) efficacy against MBL-positive bacteria in vitro, and in mice infection model, importantly, also slows down the development of higher-level resistance in NDM-1-positive bacteria. This study demonstrates a high potential of Bi(III) compounds as the first broad-spectrum B1 MBL inhibitors to treat MBL-positive bacterial infection in conjunction with existing carbapenems.

[1] Department of Chemistry, The University of Hong Kong, Pokfulam Road, Pok Fu Lam, Hong Kong. [2] Department of Microbiology, The University of Hong Kong, Sassoon Road, Pok Fu Lam, Hong Kong. [3] The Research Centre of Infection and Immunology, Li Ka Shing Faculty of Medicine, The University of Hong Kong, Pok Fu Lam, Hong Kong. [4] Department of Biology, Guangdong Provincial Key Laboratory of Cell Microenvironment and Disease Research, Shenzhen Key Laboratory of Cell Microenvironment, Southern University of Science and Technology, 518055 Shenzhen, China. [5] State Key Laboratory of Emerging Infectious Diseases, The University of Hong Kong, Sassoon Road, Pok Fu Lam, Hong Kong. Runming Wang, Tsz-Pui Lai, Peng Gao, and Hongmin Zhang contributed equally to this work. Correspondence and requests for materials should be addressed to H.S. (email: hsun@hku.hk)

Antimicrobial resistance (AMR) poses a huge threat to public health worldwide. Carbapenem-resistant Enterobacteriaceae (CRE) have been categorized as one of the urgent threats by the Centers for Diseases Control and Prevention (CDC) and kill almost half of in-patients who get bloodstream infections from these bacteria[1]. As one of the resistance determinants, metallo-β-lactamases (MBLs), including imipenemases (IMPs), Verona integron-encoded metallo-β-lactamases (VIMs), and more recent New Delhi metallo-β-lactamases (NDMs), are Zn(II)-containing enzymes that activate a nucleophilic water to cleave the β-lactam ring, conferring onto bacteria the resistance to the last resort carbapenems and all other bicyclic β-lactams that are currently used[2–4]. In particular, NDM-1-positive bacteria cause various types of infections and have spread globally now in more than 70 countries[5] since its first detection in 2009[6–8]. The geographic dissemination of NDM-1-related resistance raises great consternation because this resistance is highly transferable among many prevalent human pathogens, including Enterobacteriaceae, *Pseudomonas* spp, and *Acinetobacter* spp[9], and is usually accompanied by genes encoding other resistance determinants[10,11] or even other carbapenemases[12] in those organisms, arming them with multiple resistance to almost all classes of antibiotics available. Currently, few therapeutic options are available to treat infection from these so-called "superbugs"[13].

Combination therapy comprising an available antibiotic and a nonantibiotic that is usually an inhibitor of β-lactamase has been considered as a more economical and effective alternative than development of monotherapy with new antibiotics[2,13–17]. Such combination therapies are currently used clinically to treat infection from serine-β-lactamases (SBLs)-positive bacteria e.g., Augmentin® (amoxicillin/clavulanate), Zosyn® (piperacillin/tazobactam), and recent Avycaz® (ceftazidime/avibactam). However, up till now, no equivalent therapy is available for MBLs-positive bacterial infections, as no MBL inhibitor has been clinically approved and all clinically used SBL inhibitors are not effective toward MBLs[18]. It remains a significant challenge to design MBL inhibitors, particularly broad-spectrum MBL inhibitors, given the structural diversity and mechanistic complexities of MBLs[14]. Though chelation agents such as ethylenediamine-*N*,*N*,*N'*,*N'*-tetraacetate (EDTA) and aspergillomarasmine A[19] significantly inhibit MBLs, such indiscriminate chelating of Zn(II) might result in untoward effects on natural or endogenous metalloproteins that are critical for many biological functions[18]. Currently, most organic molecule-based MBL inhibitors, including pyridine dicarboxylates, thiol-derivatives, natural products, and even carbapenem analogs, are designed by mimicking the substrates of specific types of MBLs, and have been shown to reduce MBL activity of specific types both in vitro and in animal models[14,19–24]. However, it still remains questionable whether or when these inhibitors can be approved for clinical usage. Moreover, these inhibitors may readily encounter microbial resistance owing to rapid evolution of MBLs[21,25]. Therefore, more effective therapeutic strategies are urgently needed.

Metal compounds have been used as antimicrobial agents for centuries[26]. Auranofin (a gold antirheumatic agent) exerts broad-spectrum bactericidal activity via inhibition of bacterial thioredoxin reductase[27]. Silver nitrate and gallium nitrate enhance antibiotic activity against (multidrug-) resistant bacteria as antibiotic adjuvants[28,29]. Bismuth (Bi(III)) compounds such as colloidal bismuth subcitrate (CBS, De-Nol®) and ranitidine bismuth citrate (RBC, Pylorid®) are clinically used drugs in combination with antibiotics to treat *Helicobacter pylori* (*H. pylori*)-associated infections even for clarithromycin- and metronidazole-resistant strains, and confer no resistance even after usage for a long period of time[30,31]. Thus, reuse of metal compounds has

been considered as a promising alternative to deal with current crisis on AMR[32]. Considering that MBLs are zinc (Zn(II))-containing enzymes, the use of metal compounds to inactivate their activity may represent a new approach for the discovery of MBL inhibitors.

Here, by screening a panel of metal compounds, we show that an anti-ulcer drug, CBS, and related Bi(III) compounds exhibit potent activity toward inhibition of NDM-1, both by cell-based and enzyme-based assays. The combined use of CBS with a β-lactam antibiotic not only restores antibiotic activity, but also significantly reduces the NDM-1 evolution in NDM-1-positive *Escherichia coli* (*E. coli*). Moreover, CBS boosts the in vivo efficacy of β-lactam in murine peritonitis models. Crystallographic and biophysical studies demonstrate that Bi(III) compounds irreversibly inactivate NDM-1 by a unique mechanism, with one Bi(III) replacing two zinc ions in the active site. Cysteine (Cys208) at the active site is shown to play a pivotal role for the activity of Bi(III) compounds. Our studies strongly suggest that Bi(III) drugs or compounds might be repositioned or developed as the first broad-spectrum inhibitors of MBLs, in particular, class B1 MBLs. This study provides a new opportunity to design effective broad-spectrum MBL inhibitors for the treatment of infection caused by MBL-positive bacteria together with β-lactam antibiotics.

## Results

**Bi(III) exerts anti-MBL activity in vitro**. The antimicrobial activities of metal compounds with a β-lactam antibiotic were initially screened against a clinical isolate of NDM-1-positive *E. coli*, hereafter denoted as NDM-HK (Supplementary Table 1)[33]. The NDM-1-producing strain was confirmed to be resistant to a carbapenem, meropenem (MER) as judged by its minimal inhibitory concentration (MIC) of $16\,\mu g\,mL^{-1}$, which is higher than the breakpoint value ($>8\,\mu g\,mL^{-1}$) for MER against Enterobacteriaceae defined by European Committee on Antimicrobial Susceptibility Testing (EUCAST)[34]. The growth inhibition was examined in the presence of 10, 50, and 200 μM of a panel of metal compounds and MER at subinhibitory concentration (½MIC) for 24 h after bacterial inoculation. Primary screening gave rise to two active metal compounds i.e., bismuth nitrate (Bi(NIT)₃) and gallium nitrate with >90% growth inhibition based on optical density (OD) reading (Supplementary Fig. 1a). (Bi(NIT)₃) showed higher bioactivity than gallium nitrate in the subsequent colony-forming unit (CFU) counting, in which, surprisingly, no colony was observed in agar plate after serial dilutions when (Bi(NIT)₃) was used (Supplementary Fig. 1b). This implies that Bi(III) compounds may facilitate MER to kill NDM-1 producer. Given its clinical usage for the treatment of *H. pylori* infection, CBS was selected to further evaluate whether it can be repurposed to treat MBL-positive bacterial infection together with β-lactam antibiotics.

We first examined whether CBS (Fig. 1a) can resensitize NDM-1 producers toward carbapenem using MER as an example. Standard checkerboard microdilution method was used to monitor the interaction between MER and CBS against NDM-HK. A strain cured of pNDM-HK served as NDM-1-negative control, denoted as NDM-HK PCV (plasmid cured variant), with MIC of $0.03\,\mu g\,mL^{-1}$ for MER. CBS itself showed no or minor growth inhibition toward either NDM-1-positive (Fig. 1b) or negative bacteria (Fig. 1c) even at $256\,\mu g\,mL^{-1}$; however, when MER and CBS were used in combination, the MIC values of MER against NDM-HK gradually dropped to $2\,\mu g\,mL^{-1}$ (Fig. 1b and Table 1), which is the empirical susceptible level according to EUCAST[34], and the fractional inhibitory concentration index (FICI) was determined to be 0.250 (Supplementary Table 2), indicative of the synergistic interaction between them (FICI ≤0.5

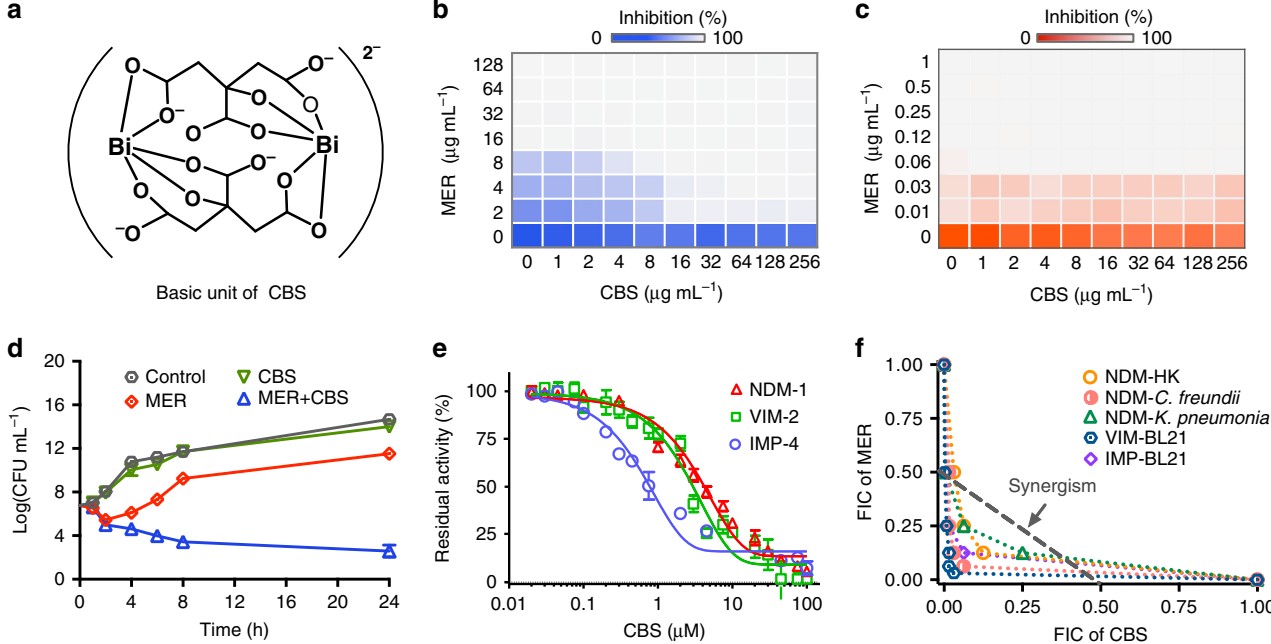

**Fig. 1** CBS inhibits the in vitro activity of MBLs. **a** Basic dimeric unit of CBS. **b, c** Representative heat plots of microdilution checkerboard assay for the combination of MER and CBS against NDM-HK (**b**) and NDM-HK PCV (**c**). **d** Time kill curves for MER and CBS monotherapy and combination therapy against NDM-HK during 24 h incubation. The concentrations of MER and CBS are 24 μg mL$^{-1}$ and 64 μg mL$^{-1}$, respectively. **e** Inhibition profiles for MBLs by CBS with IC$_{50}$ values of 2.81, 3.54, and 0.70 μM for NDM-1, VIM-2, and IMP-4, respectively. Mean value of three replicates are shown and error bars indicate the standard deviation (SD). **f** Isobolograms of the combination of MER and CBS against different MBL-positive bacterial strains. The gray line indicates ideal isobole, where drugs act additively and independently. Data overlapping with this line indicate additive effects. Data points below this line indicate synergism

**Table 1 Antibacterial activity of MER with Bi(III) compounds against different bacteria**

| Strain | MIC/MBC (μg mL$^{-1}$) of MER | | | | |
|---|---|---|---|---|---|
| | | In combination with compound at 32 μg mL$^{-1}$ | | | |
| | Alone | CBS | (Bi(NIT)$_3$) | (Bi(NAC)$_3$) | (Bi(PCM)$_2$) |
| NDM-HK (NDM-1$^+$) | 16/16 | 2/4 | 2/4 | 0.5/0.5 | 2/2 |
| NDM-*K. pneumonia* (NDM-1$^+$) | 16/16 | 4/8 | 1/1 | 1/2 | 2/8 |
| NDM-*C. freundii* (NDM-1$^+$) | 8/8 | 0.5/1 | 2/4 | 0.5/0.5 | 2/2 |
| VIM-BL21 (VIM-2$^+$) | 32/32 | 0.5/1 | 1/2 | 0.5/1 | 1/2 |
| IMP-BL21 (IMP-4$^+$) | 32/32 | 4/8 | 4/8 | 2/2 | 1/2 |

is defined as synergism). By contrast, no such synergism was detected in NDM-HK PCV (FICI = 2 as shown in Fig. 1c). The potent synergy is also demonstrated by time kill curves, which show that the population of NDM-HK at the exponential phase is significantly lowered (by more than 1000-fold) upon exposure to the drug combination of MER and CBS for 24 h (Fig. 1d). CBS could reduce MIC values of MER toward NDM-1-positive, but not negative strain, suggesting that CBS might abolish the enzymatic activity of NDM-1.

To examine whether CBS is able to inhibit enzymatic activity of NDM-1, we carried out the steady-state kinetics using MER as a substrate and observed dose-dependent inhibition on the activities of NDM-1 and related MBLs, e.g., VIM-2 and IMP-4 by CBS (Fig. 1e), ultimately leading to the activities of MBLs being inhibited by approximately 90%. Comparable half-maximum inhibitory concentration (IC$_{50}$) values of CBS were observed for NDM-1, VIM-2, and IMP-4, which were determined to be 2.81, 3.54, and 0.70 μM, respectively (Fig. 1e and Table 2). In

**Table 2 IC$_{50}$ values of Bi(III) compounds against different MBLs$^a$**

| MBL | IC$_{50}$ (μM) | | | |
|---|---|---|---|---|
| | CBS | Bi(NAC)$_3$ | Bi(NIT)$_3$ | Bi(PCM)$_2$ |
| NDM-1 | 2.81 ± 0.34 | 1.65 ± 0.60 | 0.70 ± 0.13 | 2.80 ± 0.11 |
| VIM-2 | 3.55 ± 0.78 | 2.65 ± 0.57 | 1.20 ± 0.13 | 3.71 ± 1.19 |
| IMP-4 | 0.70 ± 0.08 | 2.80 ± 0.79 | 0.99 ± 0.19 | 1.00 ± 0.37 |

$^a$Mean value of three replicates are shown and error bars indicate the standard deviation (SD)

consistence with cell-based studies, the results of the enzymatic assays confirm the capability of CBS in restoring MER antimicrobial activity was attributable to its inhibition on NDM-1 activity. We also observed a similar synergism pattern

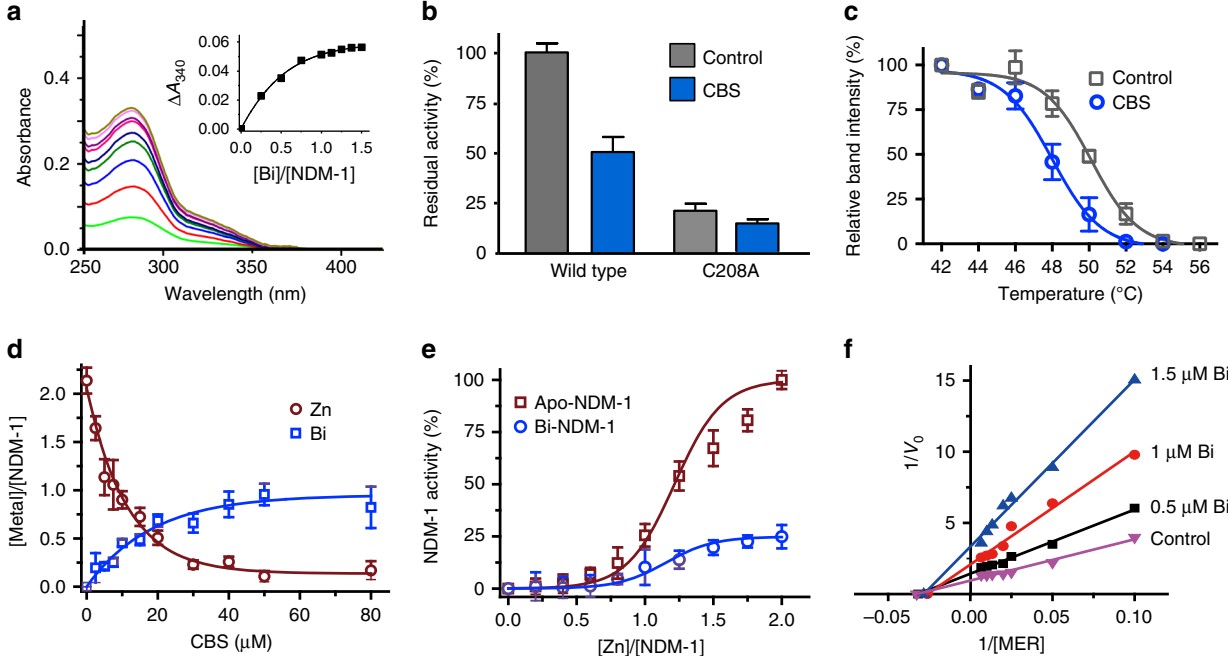

**Fig. 2** Bi(III) compounds inhibit the activity of MBLs via a unique metal replacement mechanism. **a** Different UV-vis spectra of apo-NDM-1 upon addition of 0.2–1.5 molar equivalents of Bi(NTA). The inset shows the changes of absorbance at 340 nm. **b** Normalized residual activity of wild-type (WT) NDM-1 and a variant of NDM-1-C208A in the absence or presence of CBS at $IC_{50}$. **c** Cellular thermal shift assays showing the binding of Bi(III) to NDM-1 in *E. coli* as judged from the shift of NDM-1 melting temperature from 50.1 °C to 48 °C for control and CBS-treated group, respectively. Mean value of three replicates are shown and error bars indicate SD. **d** The substitution of Zn(II) in NDM-1 by CBS as determined by ICP-MS. **e** Restoration of activity of NDM-1 upon supplementation of various ratios of Zn(II) to apo-NDM-1 and Bi-NDM-1. Mean value of three replicates are shown and error bars indicate SD. **f** The Lineweaver Burk plot shows that Bi(III) (as Bi(NIT)$_3$) inhibited NDM-1 via either a non-competitive or an irreversible inhibition mode

of the combination of CBS and MER in checkerboard assay against a panel of enterobacterial strains i.e., *E. coli*, *Klebsiella pneumonia* (NDM-*K. pneumonia*), and *Citrobacter freundii* (NDM-*C. freundii*) that harbor different MBLs (Fig. 1f and Supplementary Table 2). Collectively, we show that CBS is a potential broad-spectrum inhibitor of B1 MBLs regardless of their (sub)types or the Enterobacteriaceae that produce them.

To search for Bi(III) compounds that exhibit higher activity toward inhibition of MBLs, we synthesized a series of related Bi(III) compounds (Supplementary Fig. 2) and examined their inhibitory activities by cell-based synergy assays. Our results demonstrate that all tested Bi(III) compounds synergized with MER as adjudged by the FICI values ranging from 0.047 to 0.375, and the MIC of MER decreased by 8–64-fold (Supplementary Fig. 3a and Supplementary Table 2). Four typical Bi(III) compounds were shown in Table 1 to exemplify their potent synergy with MER against different MBL producers. In particular, a complex of bismuth with *N*-acetyl-cysteine (Bi(NAC)$_3$) exhibited the most potent inhibitory activity against different MBL-positive strains with a 32-fold, 64-fold, and 16-fold decrease in the MIC of MER against NDM-1-positive, VIM-2-positive, and IMP-4-positive *E. coli*, respectively, and the FICI values ranged from 0.063 to 0.188 (Supplementary Fig. 3b). The combination of MER and Bi(NAC)$_3$ was able to eradicate the bacterium completely within 6 h (Supplementary Fig. 4a). Such an excellent inhibitory activity may arise from cooperative inhibition of Bi(III) ions and NAC ligand dissociated from Bi(NAC)$_3$, as well as elevated cellular uptake of Bi(III) (Supplementary Fig. 4b).

Moreover, the bactericidal activity of MER against MBL-positive bacterial strains was also boosted by Bi(III) compounds, manifested by a 2–16-fold decrease in its minimal bactericidal concentration (MBC) of MER (Table 1 and Supplementary Table 2). The enzyme-based assay further demonstrates that the

Bi(III) compounds were able to inhibit MBLs with comparable $IC_{50}$ values (Table 2). Among them, (Bi(NIT)$_3$) exhibited the most potent inhibitory effect with $IC_{50}$ of 0.70 μM for NDM-1 and 1.20 μM for VIM-2, while CBS outperformed other compounds for the inhibition of IMP-4 with $IC_{50}$ of 0.70 μM. The result shows that the inhibitory effect on MBLs was generally observed for Bi(III) compounds. This is sufficient to demonstrate that the inactivation of these MBLs is attributed primarily to Bi (III) ions.

**Bi(III) inactivates MBLs via metal displacement mechanism.** We then investigated how Bi(III) compounds exerted their inhibitory activity. As shown in Fig. 2a, addition of Bi(III) (as Bi (NTA)) to apo-NDM-1 led to the appearance of an absorption band at 340 nm, which is characteristic for Bi–S ligand-to-metal charge transfer (LMCT) band. Its intensities increased, and then leveled off at a molar ratio of [Bi]/[protein] of 1, suggesting that each NDM-1 bound to one Bi(III) and a cysteine residue was involved in the binding. By fitting the titration curve with the Ryan–Weber nonlinear equation[35], $K_d$ was calculated to be 10.95 μM. Taking into account of the binding between Bi(III) and NTA (log $K_a$ = 17.55), the apparent dissociation constant ($K_d'$) was calculated to be $3.09 \times 10^{-17}$ μM. As there is only one cysteine (Cys208) at the active site, site-directed mutation of Cys208 to an alanine (as C208A) abolished the enzymatic activity by ~80% under identical conditions, confirming the importance of this residue for the enzymatic activity[36]. Accordingly, CBS at $IC_{50}$ concentration exhibited much reduced inhibitory effect on NDM-1-C208A (Fig. 2b). This phenomenon was also noted in the susceptibility test that MIC of MER against NDM-1-C208A producer remained unchanged even in the presence of 256 μg mL$^{-1}$ CBS (Supplementary Table 3). In consistence with this, the binding capacity of NDM-1-C208A to Bi(III) was reduced

significantly with only *ca.* 0.25 molar equivalents of Bi(III) bound to the mutant as revealed by inductively coupled plasma mass spectrometry (ICP-MS). These results demonstrate that Cys208 in the active site plays a pivotal role on Bi(III) inhibition of NDM-1 activity. By using cellular thermal shift assay, a method to examine the changes in thermal stabilization of targeted protein upon drug binding[37,38], we found that supplementation of Bi(III) compounds to the NDM-1-positive bacterial cells led to similar shifts in the protein-melting temperatures ($\Delta T_m$) by *ca.* 2 °C (Fig. 2c and Supplementary Fig. 5a), while the melting temperature remained almost unchanged upon treatment of bismuth compounds to the NDM-1-C208A mutant strain (Supplementary Fig. 5b), indicative of the importance of Cys208 for the binding of Bi(III) to NDM-1 in cells.

We next examined whether binding of Bi(III) induces Zn(II) release from NDM-1 by ICP-MS. As shown in Fig. 2d, the addition of increasing concentrations of CBS to native Zn-bound NDM-1 resulted in gradual decreases in the stoichiometry of Zn(II) to NDM-1, accompanied by the increase in the binding stoichiometry of Bi(III) to NDM-1, eventually ca. 2.13 molar equivalents of Zn(II) were displaced, whereas 0.96 molar equivalents of Bi(III) bound to the enzyme. Such a phenomenon is unprecedented, as in most cases, one Bi(III) ion can only displace one Zn(II) ion from an enzyme or a protein[39,40]. Significantly, inhibition of NDM-1 activity by Bi(III) was essentially irreversible as supplementation of 2 molar equivalents of Zn(II) led to only about 20% activity being restored for Bi-bound NDM-1 owing to the less extent of Bi(III) being replaced by Zn(II) from the enzyme (Supplementary Fig. 6), in contrast to full recovery for apo-NDM-1 (Fig. 2e). Limited proteolysis of purified soluble apo-bound, Zn-bound, or Bi-bound NDM-1 was also performed to explore the impact of Bi(III) on the protein stability of NDM-1. We found that the Bi-bound NDM-1 as well as apo-bound NDM-1 was readily degraded (Supplementary Fig. 7), in contrast to Zn-bound NDM-1, which resisted proteolysis by proteinase K[41]. Subsequently, we examined the change in enzyme kinetics of NDM-1 by varying the Bi(III) concentration. The addition of Bi(III) resulted in a decrease of the apparent $V_{max}$ from 0.900 to 0.365 µM s$^{-1}$ when Bi(III) concentration increased from 0 to 1.5 µM. The relevant Lineweaver Burk plot indicates either a typical non-competitive or an irreversible inhibition (Fig. 2f).

**Crystallography reveals the binding mode of Bi(III) to NDM-1.** The molecular mechanism of inhibition of MBLs by Bi(III) compounds was further investigated by X-ray crystallography. Attempts on direct co-crystallization of Bi-bound form of NDM-1 were not successful due to the amorphous precipitant that readily formed under virtually all of the crystallization conditions examined, and thus crystal soaking with CBS was used. We first crystallized and determined the high-resolution native NDM-1 structure at 0.95 Å (Supplementary Fig. 8), the highest resolution obtained so far. To facilitate the diffusion of Bi(III) ions into the crystal lattice, Zn(II) ions from the native NDM-1 crystals were removed by immersing the crystals in a pool of cryo-solution containing the metal-chelating agent, ethylenediaminetetraacetate (EDTA). The binding of Bi(III) to the protein was confirmed by X-ray excitation spectrum, which showed only the excitation peak for bismuth at around 10.8 keV, but not for the zinc peak at around 8.6 keV (Supplementary Fig. 9). Two data sets were collected for each Bi-bound crystal, one at the peak position (0.92 Å) and the other at the remote position (0.93 Å) of bismuth absorption edge. Both data sets showed clear anomalous peaks at the active site of NDM-1 with the anomalous peak for the data set collected at 0.92 Å being more significant, indicative of the

presence of Bi(III) in the crystal lattice. As shown in Fig. 3a, no significant overall conformational changes were observed between the Bi-bound and Zn-bound NDM-1 (rmsd of 0.18 Å over all $C_a$), while only subtle conformational changes were observed at the active site (Fig. 3b–d). Structural refinement using the anomalous signal showed that Bi(III) exhibited two alternative conformations at the active site. The Bi(III) is located between the two Zn(II) ions slightly closer to the Zn1 site of native NDM-1 structure as the major conformation with an occupancy of 0.5. Apart from residues that coordinate to Zn1 (His120 and His189), Cys208 as well as Asp124 that originally bind to Zn2 are also involved in Bi(III) coordination, together with a water molecule forming a trigonal prismatic geometry (Fig. 3b). Bi(III) is located slightly near the Zn2 site of native NDM-1 structure in the minor conformation with much less tendency (an occupancy of only 0.1) (Supplementary Fig. 10), for which Bi(III) coordinates to His250, Cys208, Asp124, and a water molecule, forming a distorted tetrahedral geometry (Supplementary Table 4). The bond lengths of Bi(III) with the side chains of amino acids are overall longer than those for Zn(II) in general (Fig. 3 and Supplementary Table 5), in consistence with the larger ionic radius of Bi(III) (1.03 Å) than Zn(II) (0.74 Å). The crystallographic data are in agreement with our biophysical characterization that one Bi(III) ion replaces two Zn(II), and Cys208 is crucial for Bi(III) binding to NDM-1.

**CBS suppresses resistance evolution of NDM-1.** In spite of clinical application for the treatment of *H. pylori*-associated diseases, bismuth drugs such as CBS (De-Nol®) and RBC (Pylorid®) confer no resistance even after use for a long period of time. Importantly, the combination of bismuth drugs with antibiotics significantly increases the eradication rate of drug-resistant *H. pylori*[31,42]. This is contributable to its capability of targeting multiple biological pathways[43,44], a typical feature of metallo-agents. Such unique properties might endow the potency of Bi(III) compounds to cope with resistance issue. To assess whether Bi(III) compounds exhibit such a resistance-proof characteristic in MBL-positive bacteria i.e., NDM-HK, mutant prevention concentration (MPC) was determined for MER in the absence and presence of different concentrations of CBS. We found that MER alone was unable to kill all high-level resistant mutants even at 16-fold MIC (MPC = 32 folds of MIC). In contrast, with the increase in CBS concentration, the number of mutant colonies declined significantly as shown in the heat map (Fig. 4a). The observed mutation frequency ranged from $1.31 \times 10^{-7}$ to $9.84 \times 10^{-10}$ (Supplementary Table 6). However, no such reduction in mutant colonies was noted for the NDM-1-negative strain, NDM-HK PCV (Fig. 4b). The MER MPC was lowered to 2-fold MIC against NDM-HK when ≥128 µg mL$^{-1}$ CBS or ≥64 µg mL$^{-1}$ Bi(NAC)$_3$ were used (Fig. 4c left bars and Supplementary Fig. 11b). In contrast, less mutation prevention was observed for CBS in NDM-HK PCV (Fig. 4c, right bars). Although hyper-production of NDM-1 was noted (as shown in the inset of Fig. 4d), the combination therapy significantly suppressed the evolution of high-level resistance over a period of 20 passages of NDM-HK (Fig. 4d). No such phenomenon was observed for NDM-HK PCV, implying that mutation prevention of NDM-1-positive bacteria by Bi(III) compounds is attributed to the inhibition of NDM-1 enzyme. Indeed, a MIC of 32 µg mL$^{-1}$ in the presence of 256 µg mL$^{-1}$ CBS vs 512 µg mL$^{-1}$ in the absence of CBS was found for MER against NDM-1-overexpressing *E. coli* Rosetta cells (NDM-Rosetta OX in Supplementary Table 3), confirming the ability of CBS to restore antibiotic activity against bacteria with hyper-production of NDM-1. Considering the low toxicity of CBS in humans and the acceptable mutation

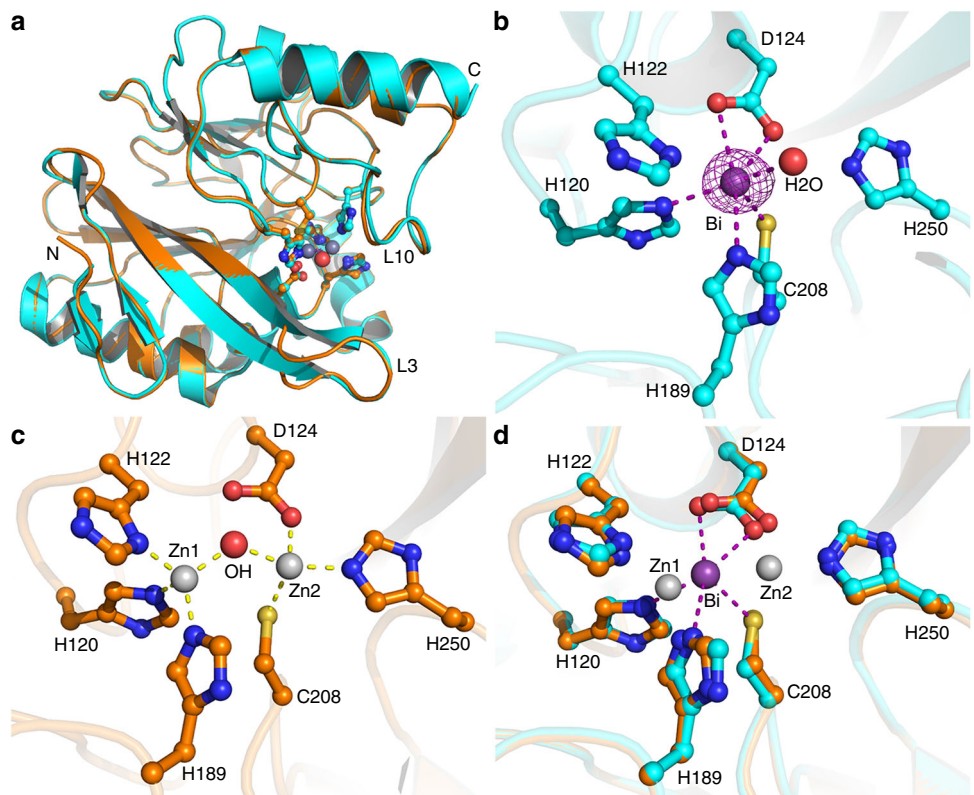

**Fig. 3** Crystallographic analysis reveals the binding mode of Bi(III) in the active site of NDM-1 **a** Superimposition of Bi-bound NDM-1 (cyan) with native Zn-bound NDM-1 (orange). Structural alignment was done over $C_\alpha$ residues using SSM algorithm in Coot and the images were generated using PyMOL. The two structures can be superimposed well with a rmsd value of 0.18 Å. **b** The active site of Bi-bound NDM-1 with the anomalous density peak of Bi shown in purple mesh contoured at 15.0$\sigma$. **c** The active site of native Zn-bound NDM-1 with two Zn(II) ions shown as gray spheres and the bridging hydroxyl nucleophile as a red sphere. **d** An overlay image comparing the relative position of Bi(III) (purple sphere) with the two Zn(II) ions (gray spheres). Bi(III) is located in between the two Zn(II) ions slightly closer to $Zn_1$

frequency, the combination of an antibiotic and CBS should have potential therapeutic applications.

**CBS restores carbapenem efficacy in vivo**. We first investigated the feasibility of CBS in restoring the antimicrobial activity of β-lactam antibiotics to treat bacterial infection in a mammalian cell culture infection model. Madin–Darby canine kidney (MDCK) epithelial cells were infected with NDM-HK at multiplicity of infection (MOI) of 200 for 5 h. The infected MDCK cells were exposed to CBS or MER monotherapy, or combination therapy overnight and lysed subsequently to enumerate the bacteria attached, penetrated, or transcytosed therein by agar plating. The results show the viable bacterial loads were still at a level of $10^6$ CFU, even when MER at 2-fold MIC was used. However, it dropped greatly to a level of $10^4$ CFU when MER and CBS were used in combination (Fig. 4e). In addition, more than 100-fold decrease in the bacterial load was also observed in the bacteria-invaded model (Supplementary Fig. 12).

We then evaluated the in vivo efficacy of the drug combination in rescuing mice from the bacterium-induced mortality using peritonitis models. Female BALB/c mice were infected systemically with a lethal dose of NDM-HK in the presence of 2% mucin. Mucin is the macromolecular component of mammalian mucus, and commixture of mucin is used to enhance the bacterial pathogenicity in mouse peritoneal cavity[28,45], whereas mice showed no morbidity or mortality when mucin was used alone. The infected mice were then administrated with a vehicle control, MER or CBS monotherapy or combination therapy, 4 h post infection, followed by twice-daily treatment via i.p. injection

(Fig. 4f). For the monotherapy, CBS failed to protect any of the mice from death within 48 h. Half of the mice in MER group also died the next day post infection, and at the endpoint of the experiment MER rescued two of the eight mice. However, in a pre-experiment, MER at an identical dose cured all the mice infected with a lethal dose of NDM-HK PCV (Supplementary Fig. 13). In contrast, the combination therapy effectively postponed the death of mice and led to an increase in mice survival rate to 50%. Additionally, no bacterial clots were observed by direct visual inspection of the infection site in the dissected mice receiving combination therapy (Supplementary Fig. 14). Similar results were obtained in a murine peritonitis model without mucin, in which 40% mice survived for up to 86 h when MER and CBS were co-administrated (Supplementary Fig. 15). Taken together, we demonstrate that the in vitro antimicrobial activity of the combination of MER with CBS could be effectively converted into in vivo efficacy.

## Discussion

Carbapenems are the antibiotic class of choice for the treatment of serious infections caused by Enterobacteriaceae producing AmpC type enzymes or extended-spectrum β-lactamases[46]. Their clinical utility is, however, greatly challenged by the emergence of carbapenemase, i.e., SBLs and more difficult-to-treat MBLs, which have extremely broad substrate spectrum on almost all β-lactams (except for aztreonam[47]), as well as being resistant to all commercially available inhibitors of SBLs. In particular, NDM-1 is one of the most widespread and threatening MBLs that binds and hydrolyzes almost all β-lactam antibiotics with Zn(II) as a

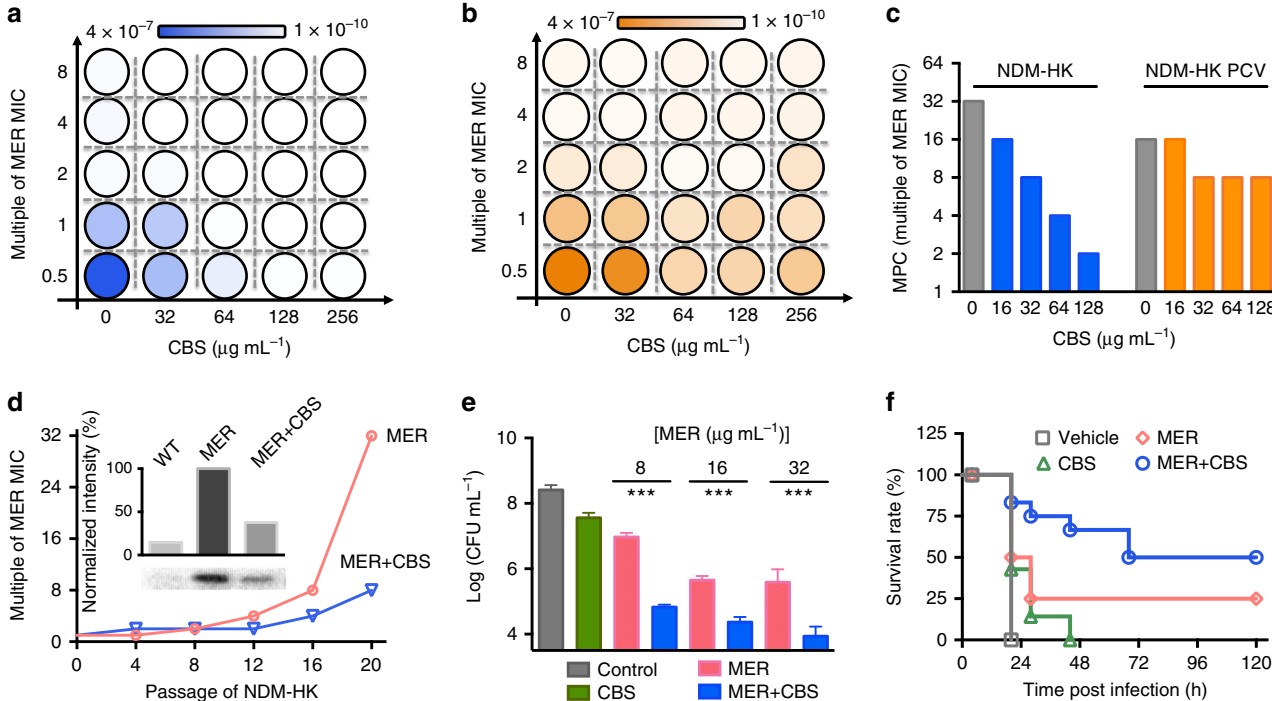

**Fig. 4** CBS suppresses the evolution of NDM-1 and boosts the antimicrobial activity of MER for the treatment of NDM-1-positive bacterial infection. **a**, **b** Heat plot visualizing the mutation frequency of (**a**) NDM-HK and (**b**) NDM-HK PCV exposed to MER in the presence of increasing concentrations of CBS. **c** Bar chart showing MPC values of MER in the presence of increasing concentration of CBS against NDM-HK and NDM-HK PCV. **d** Resistance acquisition curves during serial passage with the subinhibitory concentration of MER or combination of MER and CBS against NDM-HK. MIC test was performed every four passages. The inset shows the normalized expression level (by Western blot) of NDM-1 in the WT NDM-HK, 20th passage of NDM-HK selected by MER or by combination of MER and CBS. Original western blots is shown in Supplementary Figure 10. **e** Bar chart showing associated-bacterial load in the in vitro infection model. The concentrations ($\mu g\,mL^{-1}$) used are 8, 16, and $32\,\mu g\,mL^{-1}$ for MER and $32\,\mu g\,mL^{-1}$ for CBS. **f** Survival curves showing efficacies in a murine peritonitis infection model with the use of mucin. BALB/c mice were infected by a lethal dose of NDM-HK via intraperitoneal injection. Four groups of mice were treated with vehicle control, monotherapy of MER ($5\,mg\,kg^{-1}$), CBS ($20\,mg\,kg^{-1}$), or combination therapy of MER and CBS. $P < 0.001$, Mantel–Cox test, significant difference from the vehicle control. Eight mice per group were used in vehicle control, monotherapy of MER, or CBS and 12 mice per group in the combination therapy

cofactor. Therefore, no effective therapy is available currently. Other classes of antibiotics such as colistin and fosfomycin though exhibit antimicrobial activity against MBL-positive bacteria, usually suffer severe side effects or limitations in drug delivery[48], which further restrains the medication therapy against MBL producers. Thus, an alternative strategy is needed to develop more effective MBL inhibitors.

Herein, we demonstrate the potential of repositioning an anti-*H. pylori* drug CBS (Fig. 1a), as a new class of MBL inhibitors, as bismuth drugs such as CBS in combination with antibiotics have been recommended as first-line treatment for *H. pylori* infection; they can even overcome clarithromycin and metronidazole resistance[31]. Different from other heavy metal ions, bismuth drugs exhibit negligible toxicity in humans, attributable to glutathione, and multidrug-resistant protein-mediated disposal in humans but not in pathogens[49]. The currently accepted therapeutic daily doses of CBS in Europe deliver approximately 480 mg equivalents of bismuth oxide ($Bi_2O_3$) for up to 8 weeks[50]. No intoxication has been documented with CBS at its recommended dosage in the treatment of peptic ulcer disease, and no other serious adverse effects have been associated with CBS[51]. Our combined data show that a battery of Bi(III) compounds including CBS inhibit NDM-1, VIM-2, and IMP-4 at micromolar levels, and excitingly exhibit a synergy upon co-treatment with MER to resistant strains as reflected by the FICI values of 0.05–0.4 and significant decreases in MICs (Tables 1 and 2). The activity of MER can be restored to the level similar to those for sensitive strains. The higher activity of Bi(NAC)$_3$ toward MBLs-

positive bacteria may provide a rationale to further develop more potent Bi(III) compounds by complexation of Bi(III) with ligands that could increase Bi(III) uptake, as well as possess high affinity toward chelation of Zn(II) at the same time.

Our biophysical characterization reveals a unique feature that one Bi(III) binds to the active site, accompanied by release of two Zn(II) cofactors (Figs. 2d, 3b, c), leading to the abolishment of MBLs activity both in vitro and in vivo. Such an unexpected phenomenon might be attributed to the distinct properties of Bi (III), i.e., relative large size and high coordination numbers. Moreover, the active site of NDM-1 is located in a shallow groove encompassed by several loops with Zn(II) coordinating residues[52], which confers conformational flexibility of these residues for efficient substrate binding and catalysis turnover. In consistence with thiolphilic nature of bismuth[44], cysteine (Cys208) is critical for binding of Bi(III) as revealed by mutagenesis studies (e.g., Cys208 to Ala208), as well as the crystal structure which shows that Cys208 is involved in binding of Bi(III) in both conformations. Most organic MBL inhibitors inactivate the enzyme through binding to the substrate site of specific type of MBLs[19,22], such inhibitors are unlikely to be developed as broad-spectrum inhibitors of MBLs given the structural diversity of the enzymes at the active sites. In contrast, Bi(III) compounds inactivate the enzyme through replacement of Zn(II) by Bi(III) ion via the critical residue of cysteine at the active site. Based on the unique mechanism employed by Bi(III) to inactivate NDM-1, we strongly suggest that Bi(III) compounds can serve as a broad-spectrum inhibitor for B1 class of MBLs, although only NDM-1,

VIM-2, and IMP-4 were selected as showcase studies (Table 2). As for mono-zinc (II) B2 class of MBLs, a cysteine is found in the active sites; however, it remains to be seen whether it is accessible for Bi(III) and how fast such Bi(III)-Zn(II) exchange could be given the structural rigid nature of this class of MBLs in their active sites[53]. It may warrant further investigation to see whether Bi(III) compounds also inhibit B2 and B3 classes of MBLs.

In contrast to most organic molecule-based inhibitors, which may easily encounter resistance owing to the rapid evolution of variant MBLs just as it occurred in the case of inhibitor-resistant (IR) TEM-1 and KPC-2[54,55], the combination of CBS with MER significantly suppressed the development of high-level resistance in NDM-1 producers. This may be due to the reduced use of MER in the combination and/or reduced protein levels of NDM-1 when bacterial cells were treated with CBS (Supplementary Fig. 7). Nevertheless, the MICs of MER still raised at the end of passage compared to that of original strain (control) owing to the hyperproduction of NDM-1 as confirmed by Western blot (Fig. 4d). Whether this is due to mutations in the promoter region of the gene and/or high copy numbers of plasmids carrying the *bla* gene may warrant further studies in future.

In addition, to use as an oral drug in the triple and quadruple therapies for *H. pylori*-associated infections, intravenous or intramuscular uses of Bi(III) compounds were documented almost a century ago[56,57]. Here we enlist bismuth drugs for combatting such carbapenem-resistant systemic infection in vivo. Notably, peritoneal co-administration of CBS (at a comparable dose to current FDA-approved human dose (8 mg/kg)[58]) boosts the effectiveness of MER, leading to increased mouse survival time and reduced mortality of mouse infected with highly lethal NDM-HK (Fig. 4f). However, not all mice survived eventually, possibly due to limited studies on the pharmacokinetics and pharmacodynamics of bismuth drugs under such circumstances. Nevertheless, our robust in vitro and preliminary in vivo results provide solid foundation on the development of Bi(III) or other metallodrugs as antibiotic adjuvants to treat infection caused by MBL-producing bacteria in a wider clinical context. For example, Bi(III) is known to preferentially accumulate in kidney[59], and the local high concentration of Bi(III) may render better therapeutic effect in mouse kidney or urinary tract infection models. Moreover, as an approved drug, CBS will render a more rapid and economic route into clinical trials and even to patents, if successful.

In summary, we demonstrate that CBS and related bismuth compounds are a class of novel and potent inhibitors of NDM-1 and other MBLs both in vitro and in vivo. Distinct from previously reported organic molecule-based inhibitors, Bi(III) irreversibly inhibits MBLs via a unique metal displacement mechanism, with the cysteine residues at the active site being critical for bismuth coordination. We thus anticipate that CBS or other bismuth compounds can be repositioned or developed as a new class of broad-spectrum inhibitors for B1 MBLs. Given that Bi(III) compounds are already clinically used drugs and their selective toxicity toward pathogens. Such broad-spectrum MBL inhibitors together with antibiotics as co-therapy will undoubtedly open a new horizon for the treatment of infection caused by MBL-positive bacteria. Indeed, combined use of such resistant breakers with existing antibiotics represents a new and more economical therapy to effectively solve the problem of antibiotic resistance[60].

## Methods

**Chemical reagents**. Meropenem was purchased from TCI Chemicals (Shanghai). Kanamycin sulfate, Luria-Bertani (LB) Broth Powder, and LB agar were purchased from Affymetrix. All other chemicals were purchased from Sigma-Aldrich unless otherwise stated.

**Bacteria**. The bacteria employed for cell-based and animal studies are listed in Supplementary Table 1. All the *E. coli* variants were made as described below. The NDM-HK PCV was screened by 20[th]-generation serial passages of NDM-HK in antibiotic-free medium. The loss of $bla_{NDM-1}$ gene was confirmed by both PCR and susceptibility test.

**Preparation of Bi(III) compounds**. CBS and RBC were kindly provided by Livzon Pharmaceutical Group. Bi(NIT)$_3$ (Bi(NO$_3$)·5H$_2$O) was purchased from Sigma-Aldrich. The chemical structures of all the ligands used are shown in Supplementary Fig. 2. Bi(EDTA) (EDTA: ethylenediaminetetraacetate) and Bi(NTA) (NTA: nitrilotriacetate) were prepared by mixing bismuth subcarbonate ((BiO)$_2$CO$_3$) and appropriate amounts of ligands, followed by refluxing overnight. The solution was filtered while hot and was cooled down naturally. Crystals were collected and washed by water and ethanol the next day. Bi(NAC)$_3$ (NAC: *N*-acetyl-cysteine), Bi(GSH)$_3$ (GSH: glutathione)[39], and Bi(TBC)$_2$ (TBC: tetra-bromocatechol)[61] were prepared by mixing Bi(NO$_3$)·5H$_2$O with the appropriate amounts of ligands in methanol. Any impurities were removed by filtering and the resulting solution was evaporated slowly to obtain the solid product.

Bi(TPP) (TPP: tetraphenylporphyrin) were synthesized using a modified method[39]. In brief, to 80 mL of refluxing pyridine containing 0.10 g (0.2 mmol) of TPP was added 1.99 g of Bi(NO$_3$)$_3$·5H$_2$O (2.1 mmol), and a further 2.00 g (4.1 mmol) was then added 3 h later. After further reflexing for 5 h, a large amount of pyridine was removed via rotary evaporation and the resulting thick, green mixture was then left to dry overnight under vacuum to remove residual solvents. Green solids that were obtained were then washed with chloroform and rotary evaporated to ensure that all pyridine solvent was removed. The dark-green solids were then purified on a silica gel column. The compound was purified by washing the column with first chloroform and then chloroform:methanol in a ratio of 10:1. FAB-MS$^+$ was used to characterize the complex with *m/z* of 820.9 ([Bi(L1)]$^+$; calculated: 821.2).

Bi(CPL)$_2$ was prepared by mixing BiCl$_3$ (1 mmol) and D-captopril (3 mmol, J&K Scientific) in methanol at room temperature with constant stirring overnight until a color change from colorless to yellow had occurred. The methanol was removed under vacuum, and the resulting yellow solid product was washed successively with ethanol and water before recrystallization from methanol, yielding the product. $^1$H NMR (300 MHz, CD$_3$OD, δ ppm): 4.73 (m, 1 H), 3.83 (m, 1 H), 3.71 (s, 3 H), 3.05 (m, 1 H), 2.26 (m, 1 H), 2.03 (m, 2 H), and 1.24 (d, 3 H, *J* = 7.0 Hz). ESI-MS$^-$ *m/z*: 639.08 (100%). The peak at *m/z* 639.08 corresponds to [BiL$_2$]$^-$ (calculated: 639.51).

Bi(PCM)$_2$ was prepared by mixing BiCl$_3$ (1 mmol) and D-penicillamine (3 mmol, J&K Scientific) in methanol with stirring, resulting in an immediate color change from colorless to pale yellow. The mixture was further stirred overnight at room temperature before any remaining solid was filtered off and the solvent from the filtrate was removed under vacuum. The resulting solid was recrystallized from methanol, yielding a yellow solid product. $^1$H NMR (300 MHz, D$_2$O, δ ppm): 3.48 (s, 1 H), 0.90 (s, 3 H), and 0.83 (s, 3 H). ESI-MS$^+$ *m/z*: 1214.8 (80%), 860.2 (100), 505.3 (95), and 356.3 (50). The peaks at *m/z* 356.3, 505.3, 860.2, and 1214.8 correspond to [BiL]$^+$ (calculated: 356.01), [BiL$_2$]$^+$ (calculated: 505.06), [Bi$_2$L$_3$]$^+$ (calculated: 860.06), and [Bi$_3$L$_4$]$^+$ (calculated: 1214.46), respectively.

**DNA manipulations and construction of plasmids**. All the plasmids used as templates for PCR were purified using the gel extraction kit (QIAprep Spin Miniprep Kit (250), QIAGEN). All PCR primers were synthesized from Thermo Fisher and listed in Supplementary Table 7. PCR was performed using KOD Hot Start DNA Polymerase (Novagen) based on the reaction conditions described in the protocols by the manufacturers. All restriction enzymes were used directly (New England Biolab). The amplified genes of NDM-1, VIM-2, and IMP-4 were subsequently inserted into pET-28a plasmid with an incorporated N-terminal His-Tag using T4 DNA ligase to form the vector pET-28a-NDM-1, pET-28a-VIM-2, and pET-28a-IMP-4, respectively. Expression vector for NDM-1 variant (NDM-1-C208A) was generated by PCR using the standard protocols from the Phusion Site-Directed Mutagenesis Kit (New England Labs) with pET-28a-NDM-1 as the template. The constructed plasmids were subsequently transformed into XL1-Blue competent cells for molecular cloning.

**Protein purification**. A single colony of *E. coli* BL21(DE3) transformed with the respective MBL was inoculated in LB medium supplied with 50 μg mL$^{-1}$ kanamycin and grown at 37 °C. Protein overexpression was induced using 0.2 mM IPTG supplemented with 0.2 mM ZnSO$_4$ at OD$_{600}$ 0.6. The bacterial culture was incubated at 25 °C overnight. To purify the respective protein, the cultured cells were harvested by centrifugation at 4500 × *g* and resuspended in a lysis buffer (20 mM HEPES, 0.5 M NaCl, and 1 mM PMSF at pH 7.0). The cells were ice-cooled and lysed by sonication and then centrifuged at 35000 × *g* for 30 min to remove the majority of cell debris. The supernatant was filtered using Minisart syringe filter (0.45 μm) to remove any remaining large and insoluble cell debris, and was then applied to a 5 mL Ni(II)-loaded HiTrap chelating columns (GE Healthcare) at a rate of 2 mL min$^{-1}$. The column was washed using five column volumes of washing buffer (20 mM HEPES, 0.5 M NaCl, and 30 mM imidazole at pH 7.0). The protein was eluted out using four column volumes of the same buffer with gradient

amounts of imidazole, and was subsequently dialyzed against the cleavage buffer (20 mM HEPES, 0.15 M NaCl at pH 7.0). The N-terminal His-tag of the fusion protein was cleaved by adding 50 NIH units of thrombin at 25 °C for 3 h with mild shaking and the cleaved His-tag was separated from the resulting proteins by passing through the Ni(II)-NTA column again using washing buffer so that >90% of the proteins were in the flow-through fraction. The enzymes were further purified using HiLoad 16/60 Superdex 200 pg gel filtration column. The samples were then concentrated using Amicon Ultra-15 Centrifugal Filter Devices (Millipore) and separated into aliquots after dialysis with storage buffer (20 mM HEPES, 0.1 M NaCl at pH 7.0 for long-term storage at −80 °C.

**IC$_{50}$ enzyme inhibition assay**. Freshly prepared 50 nM of Zn$_2$-NDM-1, Zn$_2$-VIM-2, or Zn$_2$-IMP-4 in 50 mM HEPES/Na pH 7.0, 100 mM NaCl were first incubated with various concentrations of CBS for 1 h at 25 °C, then mixed with equal volume of 0.2 mM MER. The assay was performed in a 1 cm quartz cuvette using the kinetic mode on a Varian Cary50 UV-visible spectrophotometer at 25 °C. The decrease in absorbance at 300 nm due to ring-opening of MER was monitored every second for a duration of 10 min until the reaction was completed. The initial rates were extracted and calculated from each reaction curves for fitting IC$_{50}$ curves.

**UV-vis spectroscopy**. UV-vis spectra were collected on a Varian Cary 50 spectrophotometer at a rate of 360 nm min$^{-1}$ using a 1-cm quartz cuvette at ambient temperature. Aliquots of 2 mM (Bi(NTA)) stock solution were stepwise titrated into apo-NDM-1 sample (50 μM) in a titration buffer (20 mM HEPES, 50 mM NaCl at pH 7.4) and UV-vis spectra were recorded in a range of 220–600 nm at least 30 min after each addition. The binding of Bi(III) to apo-NDM-1 was monitored by the increase in absorption at 340 nm due to LMCT involving the only cysteine residue (Cys208). The UV titration curve was fitted to Ryan–Weber nonlinear equation[62] (1).

$$I = \frac{I_{max}}{2C_p}\left[(K_d + C_m + C_p) - \sqrt{(C_p + C_m + K_d)^2 - 4C_mC_p}\right],\qquad(1)$$

where $I$ stands for UV absorbance intensity; $I_{max}$ for maximal UV absorbance; $C_p$ and $C_m$ refers to the total concentrations of proteins and ligands, respectively; $K_d$ is the dissociation constant. For Bi$^{3+}$, the dissociation constant from NDM-1 was derived by $K_d' = K_d/K_a$, where $K_d$ is the dissociation constant of Bi(NTA) from NDM-1 determined from Ryan–Weber nonlinear fitting[35] and $K_a$ is the formation constant of Bi(NTA) with log $K_a = 17.55$.

**Zinc displacement analysis by ICP-MS**. To monitor the displacement of Zn(II) by Bi(III), ICP-MS was employed to accurately quantify $^{209}$Bi and $^{66}$Zn contents in various purified NDM-1 samples. Purified Zn-bound-NDM-1 (20 μM) dissolved in trace-metal-free ICP-MS buffer containing 50 mM HEPES, pH 7.0, was incubated with various concentrations of CBS at 25 °C for 5 h with mild shaking. The samples were subsequently dialyzed in ICP-MS buffer to remove unbound-metal ions and were then acidified and subsequently analyzed using an ICP-MS spectrometer (Agilent 7500a, Agilent Technologies, CA, USA) with $^{115}$In as an internal standard for $^{209}$Bi, $^{66}$Zn, and $^{34}$S contents, which are used to quantify protein concentration.

**Zinc supplementation assay**. To investigate whether bismuth inhibits the enzyme reversibly, enzyme activities of Bi-NDM-1 and apo-NDM-1 were compared upon the supplementation of Zn(II). Bi-bound NDM-1 (50 nM) was prepared by pre-incubation of apo-bound NDM-1 with CBS for 2 h at 25 °C, followed by removal of unbound Bi(III) and the bound Bi was verified by ICP-MS. The above protein solutions were mixed with ZnSO$_4$ at concentration up to 2 molar equivalents to NDM-1 and 100 μM MER. The change in absorbance at 300 nm was monitored on a Varian Cary 50 UV-visible spectrophotometer at 25 °C for calculation of reaction rates. Reaction rate of apo-NDM-1 with addition of 2 molar equivalents of ZnSO$_4$ was normalized to 1. The metal content of Bi-bound NDM-1 treated with Zn(II) was measured and analyzed by ICP-MS as described in the previous section.

**Limited proteolysis of NDM-1 protein**. Limited proteolysis[41] was performed to examine the in vitro stability of different forms of NDM-1. In brief, the aliquots (150 μg) of pure apo-bound, Zn-bound, and Bi-bound NDM-1 were treated with 2 μg of proteinase K (Fungal, Invitrogen, >20 U/mg) in 10 mM Tris, 5 mM CaCl$_2$, pH 8, at 16 °C. Aliquots were taken at various time intervals. The reaction was quenched with 5 mM PMSF, and samples were then subjected to SDS-PAGE and Coomassie blue staining. A PageRuler Prestained Protein Ladder #26616 (Thermo) was used as a standard marker.

**Michaelis–Menten kinetics**. NDM-1 (50 nM) was incubated with Bi(NIT)$_3$ (0.5, 1.0, and 1.5 μM) for 1 h at 25 °C with gentle shaking. The assay was performed in a 96-well microplate reader at 298 K. The final assay buffer contains 50 mM HEPES at pH 7.0, 100 mM NaCl, with MER as the substrate ranging from 10 to 150 μM. Control experiment was also performed in the absence of inhibitors under the same

conditions. The $K_m$ and $V_{max}$ for both the uninhibited and inhibited reactions were obtained by fitting the data into the double reciprocal Lineweaver–Burk plots.

**Detection of NDM-1 expression level by western blot**. NDM-1 protein level was determined by SDS-PAGE followed by western blot over the whole-cell lysates of different E. coli stains. Typically, each logarithmic phase culture was lysed by sonication in sonication buffer (50 mM HEPES, pH 7.3, 100 mM NaCl). Bacterial lysates were harvested by centrifugation and normalized according to total protein concentrations, as quantified by bicinchoninic acid assay (Pierce BCA Protein Assay Kit, Thermo Scientific). All the samples were resolved on a 13% SDS-PAGE gel and electrotransferred to a PVDF membrane (Hybond-P, GE Healthcare). A PageRuler Prestained Protein Ladder #26616 (Thermo) was used as a standard marker. Diluted NDM-1 monoclonal antibody (NOVUS Biologicals) and the secondary antibody goat anti-mouse immunoglobulin G (IgG)/alkaline phosphatase (AP) conjugate were applied after performing the standard blotting procedures. The NDM-1 bands were calorimetrically developed with specified ratio of substrates comprising nitroblue tetrazolium/5-bromo-4-chloro-3-indolyl phosphate (NBT/BCIP) for 15 min. The software ImageJ[63] was used to quantify the signal of each of the bands for analysis. The software GraphPad Prism (version 6.2 for Mac, GraphPad Software, La Jolla CA, USA, www.graphpad.com) was used to analyze the resulting plots.

**Cellular thermal shift assay**. The cellular thermal shift assay was performed according to a standard method[38]. Clinical isolate of NDM-1-positive E. coli (NDM-HK) was cultured overnight in the absence or presence of 100 μM Bi(III) compounds, i.e., CBS, Bi(NAC)$_3$, Bi(NIT)$_3$. The bacterial pellets were harvested and washed with PBS for 4 times. The cell suspensions were aliquoted into PCR tubes and heat treatment was performed at the designated temperature ranging from 40 °C to 70 °C for 3 min in a 96-well thermal cycler. The tubes were cooled immediately at room temperature for another 3 min after heating and the heating procedures were repeated for three cycles. For the cell lysis, the samples were frozen–thawed for two cycles in liquid nitrogen and thermal cycler set at 25 °C. The samples were vortexed gently after each cycle and were centrifuged at 20,000 × g to obtain the supernatant when the second cycle was finished. All the samples were subjected to western blot analysis for detection and quantification of NDM-1 content as described above.

**Primary screening upon different metal compounds**. Metal salts used for the screening involved bismuth nitrate (Bi(III)), gallium nitrate (Ga(III)), sodium stibogluconate (Sb(V)), chromium chloride (Cr(III)), cobalt chloride (Co(II)), nickel chloride (Ni(II)), ruthenium chloride (Ru(II)), and copper sulfate (Cu(II)). Briefly, different concentrations (10, 50, and 200 μM) of metal compounds were added to LB medium containing 8 μg mL$^{-1}$ of MER in 96-well plates. About 2 × 10$^6$ CFU mL$^{-1}$ logarithmic cultures of NDM-HK were added to each well of the plates and co-incubated for overnight. The growth inhibition of bacteria was monitored by OD reading at 600 nm and serial dilution in LB agar plate. Wells with no antibiotics or metal compounds served as growth controls, and wells with no bacteria added served as background controls. Each test was performed in triplicate. The inhibition was calculated as [1−(OD$_{sample}$−OD$_{background}$)/ (OD$_{control}$−OD$_{background}$)] ×100%. CFU was counting by 10-fold serial dilution in PBS and 10 μL was spotted in LB agar plate.

**Microdilution MIC and MBC assay**. MIC values were determined by standard broth micro-dilution method (Clinical and Laboratory Standards Institute (CLSI) M100-S20, 2010)[64]. Briefly, bacterial cells were cultured in LB broth overnight at 37 °C at 250 rpm and the OD was measured at 600 nm (OD$_{600}$). The bacterial density was adjusted to about 1 × 10$^6$ CFU mL$^{-1}$ and checked by CFU counting on agar plates afterwards. Tested antibiotics or Bi(III) compounds were added triplicately into individual wells of flat-bottomed 96-well plates and performed 2-fold serial dilution, followed by addition of prepared bacterial inocula. The plate was then incubated at 37 °C for overnight. Wells with no antibiotics or Bi(III) compounds served as growth controls and wells with no bacteria added served as background controls. The MIC was determined as the lowest concentration of a drug that could inhibit the growth of microorganism by both visual reading and OD reading at 600 nm using a microtiter plate reader.

For the test of NDM-Rosetta OX, the growth condition briefly goes as follow. Overnight culture of NDM-Rosetta was 1000-fold diluted into fresh LB medium and regrew to OD ~0.6. The overexpression of NDM-1 was induced by addition of 1 mM isopropyl β-D-1-thiogalactopyranoside (IPTG) for 4 h at 37 °C. The following susceptibility test was performed as described in the Microdilution MIC and MBC assay section in the methodology by using the resulting bacterial pellets with the supplementation of 200 μM IPTG.

For the test of C208A-Rosetta, the growth condition briefly goes as follow. Overnight culture of C208A-Rosetta was diluted into fresh LB medium (1000-fold) and regrew to OD ~0.6. The overexpression of NDM-1-C208A was induced by addition of 200 μM isopropyl β-D-1-thiogalactopyranoside (IPTG) for 18 h at 25 °C. The following susceptibility test was performed as described in the Microdilution MIC and MBC assay section in the methodology by using the resulting bacterial pellets with the supplementation of 200 μM IPTG.

At the end of MIC assay, 50 μL of aliquots of each well (containing a specified antimicrobial concentration) for each isolate tested were applied to a LB agar plate and incubated at 37 °C overnight. Resulting growth (or lack of growth) was examined after overnight culturing and the lowest concentration that inhibits 99.9% of the original culture was taken as MBC.

For drug combination test, antibiotics and Bi(III) compounds were co-added at concentrations up to eight times higher than the MIC of the drugs tested alone. Other procedures were kept strictly the same. The FICI was determined according to the following equation: FICI = $FIC_A$ + $FIC_B$ = $C_A/MIC_A$ + $C_B/MIC_B$, where $MIC_A$ and $MIC_B$ are the MIC values of compounds A and B, respectively, when functioning alone, and $C_A$ and $C_B$ are the concentrations of compounds A and B at the effective combinations. Synergism was defined when FICI ≤ 0.5, indifference was defined when FICI > 0.5 and < 4, and antagonism was defined when FICI≥4[65]. A volume of 256μg mL$^{-1}$ was set as the MIC of Bi(III) (except for Bi(NAC)$_3$) for the determination of FIC values. All of the determinations were performed at least in triplicate on different days.

**Time kill assay**. Time kill assay was used to further explore the synergy between MER and Bi(III) compounds. In a typical assay, NDM-HK was cultured overnight and diluted 1:250 into LB broth at 37 °C for 3 h to reach log phase. The initial bacterial density was adjusted to ~10$^7$ CFU mL$^{-1}$ and then exposed to MER, Bi(III) compounds either alone or in combination. LB broth with no drugs served as a control. Aliquots of bacterial suspension were withdrawn at different time intervals (0, 1, 2, 4, 6, 8, and 24 h) for inspection of bacterial viability by agar plating. The concentrations of the drugs used in the test are 24 μg mL$^{-1}$ for MER, 64 μg mL$^{-1}$ for CBS, and 32 μg mL$^{-1}$ for Bi(NAC)$_3$. Data from three independent experiments were averaged and plotted as log$_{10}$ CFU mL$^{-1}$ vs. time (h) for each time point over 24 h. All assays were triplicated and performed three times on different days.

**Measurement of bismuth uptake by ICP-MS**. Five colonies of NDM-HK were grown in LB broth to ~OD$_{600}$ 0.1. Bi(III) compounds at various concentrations (0, 1, 2, 5, 10, 20, 30, 50, 100, 200, and 500 μM) were added to the respective wells in 24-well plate in triplicate. Bacterial cell pellets were collected after 24 h incubation at 37 °C, followed by washing with PBS for six times. The harvested bacterial pellets were dissolved by 60 μL of 68% HNO$_3$ at 60 °C overnight using a Thermolyne DriBath. The dissolved samples were diluted to appropriate concentration for quantification of metals by ICP-MS (Agilent 7500a, Agilent Technologies, CA, USA) with $^{115}$In as an internal standard. Metal quantifications were triplicated and average values were used.

**X-ray crystallography**. Crystals of native NDM-1 were grown using hanging drop vapor diffusion method. The crystals were grown using precipitant containing 0.1 M Bis-Tris at pH 5.5, 15% PEG 3350 (w/v), and 20 mM L-proline. One microliter (μl) of protein solution at concentration of 50 ~ 100 mg ml$^{-1}$ was mixed with 1 μl of precipitant, sealed, and incubated at 20 °C. Diamond-like or rectangular crystals appeared within a day after seeding and grew up to full size within a week. They generally diffracted to resolutions of 0.93–1.0 Å.

NDM-1 crystals were cross-linked with 25% (v/v) glutaraldehyde at 25 °C for 30 min and then soaked in a chelating solution (0.1 M sodium acetate, pH 4.6, 25% PEG 3350, 20% glycerol, 10 mM EDTA) overnight. The crystals were washed three times in cryo-protectant solution (0.1 M Bis-Tris, pH 5.5, 25% PEG 3350, 20% glycerol) and then soaked with bismuth compounds. Soaking was done for 17 h by adding 5 mM TCEP and 1 mM bismuth compounds (bismuth nitrate or CBS). The crystals were further washed with the cryo-protectant solution for four times and flash-frozen into liquid nitrogen.

Two data sets were collected at BL17U1 at the Shanghai Synchrotron Radiation Facility (SSRF) at two specific wavelengths of 0.92 and 0.93 Å, which crossed the L$_3$ absorption edge of elemental Bi. Excitation scan was performed to further confirm the absence of zinc ions after soaking. The diffraction data were processed with HKL2000 at SSRF. Molecular replacement was performed using the program Phaser[66] from the CCP4 suite and the ampicillin-bound NDM-1 (PDB code: 3Q6X) as a searching model. Cycles of refinement with the anomalous data were done using Refmac[67] and with careful manual rebuilding in Coot[68]. The anomalous signal strength was compared between the two data sets collected at wavelength of 0.92 Å and 0.93 Å. The Bi(III) occupancy was refined based on the Bi anomalous signal at early refinement stage and was assessed by atomic B-factor in later stages. TLS refinement was incorporated into later refinement processes. Solvents were added automatically in Coot and then manually inspected and modified. The final models were analyzed with MolProbity[69]. Data collection and model refinement statistics are summarized in Supplementary Table 4.

**Resistance study**. To measure MPC[70], NDM-HK at 1–2 × 10$^{10}$ CFU was plated onto LB agar containing MER and CBS at different concentrations and incubated at 37 °C. After 48 h incubation, to any plates with observable colonies, up to four colonies were picked and re-cultured, followed by the measurement of their MIC values. Any MICs of MER that were greater than the original value were determined as higher-level resistant mutant colony. The concentration that restricted the growth of mutant colonies was determined as MPC. In an identical experiment, the higher-level resistant mutant colonies were enumerated. The relative mutation

frequency at each MIC for each strain/antibiotic pair was calculated as the proportion of resistant colonies per inoculum.

For the serial passage assay[71], an overnight culture of NDM-HK was diluted to ~10$^7$ CFU mL$^{-1}$ in LB broth. The as-diluted bacterial suspension was added to each well of 96-well plate supplemented with the drug at 0.5-fold, 0.75-fold, 1-fold, 1.25-fold, 1.5-fold, 1.75-fold, 2-fold, 3-fold, and 4-fold MIC, respectively. The drug concentrations for in vitro selection were increased to 2-fold, 4-fold, 6-fold, 8-fold, 16-fold, 24-fold, and 32-fold of MIC, respectively after 12 bacterial passages. All the plates were incubated at 37 °C and the growth of cultures was checked at 24 h intervals. Cultures from the second highest concentrations that allowed growth were performed 1:1000 dilution into fresh medium supplemented with the same concentrations of drugs. For MER, 1-fold of MIC was set as 16 μg mL$^{-1}$. For the combination of MER and CBS, 1-fold of MIC was set as 2 μg mL$^{-1}$ MER + 32 μg mL$^{-1}$ CBS. This in vitro passage was repeated for 20 days. MIC of MER was determined every four passages.

**In vitro cell infection assay**. MDCK cell was cultured in minimum essential Media (MEM) supplemented with fetal bovine serum (FBS, 10%) and grown at 37 °C in 5% CO$_2$-humidified atmosphere for 3 days. About 1.0 × 10$^5$ MDCK cells were seeded per well in 24-well plates and incubated as described above for 48 h to ensure confluency. Logarithmic cultures of NDM-HK were washed with PBS for three times and re-suspended in MEM/10% FBS resulting in the initial bacterial density of about 2.0 × 10$^7$ CFU mL$^{-1}$. Then, 500 μL of bacterial suspension were added to each well and substituted for the previous MDCK culture medium. The plates were centrifuged at 800 × g for 10 min and then incubated for another 6 h, executing the bacterial infection at multiplicity of infection (MOI) of 200. We then used two infection models for examination, viz. the cell-associated bacterial infection and cell-invaded bacterial infection. The cell-associated bacteria are herein defined as bacteria that attach to, penetrate, or transcytose in MDCK cells, while the cell-invaded bacteria are defined as bacteria that penetrate or transcytose MDCK cells. For the cell-invaded infection, the infected cells were incubated in culture medium supplemented with ciprofloxacin (100 μg mL$^{-1}$) for 1 h to remove extracellular bacteria. Then, the treated cells were washed vigorously with PBS for six times and replenished with culture medium. For the cell-associated infection, the infected cells were only washed vigorously with PBS for six times to remove unbound bacteria. The infected MDCK cells were then exposed to either MER or CBS, or their combination for overnight under identical cell culture condition. Cells in the absence of drugs served as a control. The bacterial loads were examined by lysing MDCK cells with 1% Triton X-100 in PBS and serially diluting the resulting lysates to enumerate bacterial colonies by agar plating. The assay was performed in triplicate, repeated three times, and results were expressed as average ± SD.

**Murine peritonitis infection**. All experiments were approved by, and performed in accordance with the guidelines approved by Committee on the Use of Live Animals in Teaching and Research (CULATR), the University of Hong Kong. A total of 6–8-week-old, female BALB/c mice (18–22 g) were purchased from Charles River Laboratories, Inc. and were used in all mouse studies. The animals were randomized to cages for each experiment.

In mucin-assisted model, for infection with NDM-1-positive bacteria, an overnight culture of NDM-HK was performed 1:250 dilution in 100 mL LB broth and re-grew to about OD$_{600}$ 0.3 in a 500-mL flask after 2.5 h shaking at 37 °C, 250 rpm. Bacterial pellets were collected and washed by PBS buffer three times for further use. Mice were infected intraperitoneally (i.p.) with a dose of 1 × 10$^5$ CFU of bacteria in PBS supplemented with 2% mucin. Four groups of mice were i.p. administered 4 h post infection with a 100-μL aliquot of vehicle control, monotherapy of MER (10 mg kg$^{-1}$) or CBS (10 mg kg$^{-1}$), or combination therapy (n = 8, 8, 8, and 12, respectively). Twice-daily treatment via i.p. injection was continued throughout the whole experimental course. For the infection with NDM-1-negative bacteria, all the operations of infection were similar to those used for NDM-HK, except that NDM-HK PCV was used as infection-causing bacteria. Two groups of mice were i.p. administered 4 h post-infection with a 100-μL aliquot of vehicle control and monotherapy of MER (10 mg kg$^{-1}$) (n = 5 for each group). Twice-daily treatment via i.p. injection was continued throughout the whole experimental course. Body weights and mice survival were monitored till endpoint of the experiment.

In mucin-free survival experiment, the mice were infected with a dose of 5 × 10$^7$ CFU of NDM-HK in PBS. The mice received monotherapy of MER (50 mg kg$^{-1}$) or CBS (50 mg kg$^{-1}$), or combination therapy 0.5 h post infection (n = 5 for each group). Other experimental operations and conditions were kept the same as that in mucin-assisted model. Body weights and mice survival were monitored for endpoint till endpoint of the experiment.

**Data availability**. The coordinates and structure factors for both zinc-bound native NDM-1 and Bi(III)-bound NDM-1 were deposited at Protein Data Bank with accessing code 5XP6 and 5XP9, respectively. Other data are available from the corresponding author upon reasonable request.

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

## Acknowledgements

We gratefully acknowledge the Research Grants Council of Hong Kong (703913P, 17305415P, and 17333616P), National Natural Science Foundation of China (31670753), the Guangdong Science and Technology Program (2017B030301018), the Health and Medical Research Fung (HKM-15-M10), the University of Hong Kong (for an e-SRT on Integrative Biology) and research grants from Shenzhen (JCYJ20160608140912962, ZDSYS20140509142721429) for financial support. We thank Profs. Jiandong Huang and Quan Hao (Li Ka Shing Faculty of Medicine, HKU) for helpful discussion. The crystal diffraction data were collected at Shanghai Synchrotron Radiation Facility (SSRF), the Chinese Academy of Sciences. We thank the staff at BL17U1 beamline of SSRF for their generous help.

## Author contributions

H.S., H.L., R.Y.-T.K., P.-L.H., and P.C.-Y.W. conceived idea and designed experiments; R.W. synthesized the compounds; R.W., T.-P.L., and P.G. performed the antimicrobial experiments; R.W. and P.G. performed animal experiments; H.Z., T.-P.L., and G.M. overexpressed, purified and crystallized the proteins, and solved the structures (Zn-bound NDM-1 and Bi-bound NDM-1) by X-ray crystallography; and H.L., R.W., and H.S. principally wrote the manuscript with input from all.

## Additional information

**Competing interests:** The authors declare no competing financial interests.

