## [Peer Review File · Nature Communications]

Reviewers' comments:

Reviewer #1 (Remarks to the Author):

Inhibition of Metallo-beta-lactamases has been a challenging task, with efforts spanning from chelating agents, substrate mimics, and possible transition state analogues. In this manuscript, the authors report a novel strategy, namely the use of bismuth compounds, that have been used as antimicrobial for many decades. The authors show that some Bi(III) compounds, particularly colloidal bismuth subcitrate (CBS) inhibit several B1 lactamases irreversibly by displacing the native Zn(II) ions and binding to the Cys ligand in the active site. The work shows the effect in the purified enzymes NDM-1, VIM-2 and IMP-4 and in bacteria expressing these MBLs. Then, they show that Bi(III) displaces the native metal ions and report the crystal structure of the inactive Bi(III)-substituted NDM-1. Finally, they demonstrate that CBS is able to partially restore the action of meropenem in an animal infection model.

Overall, this is an outstanding piece of work in which the authors use different, complementary approaches to support their contention.

I do have some comments that the authors may want to consider to improve the quality of the manuscript. Major comments:

1. NDM is a membrane-bound protein that is degraded upon metal deprivation (Gonzalez et al (2016), Nature Chem. Biology). A discussion of the impact of metal deprivation on this strategy would enrich the work.
2. The cellular thermal shift assay is performed in sonicated cells, and the authors measure the soluble protein. As mentioned before, NDM is a membrane-bound protein (Gonzalez et al (2016), Nature Chem. Biology). They should clarify what they are actually measuring here.
3. The occupancies of both Bi(III) conformations are 0.5 and 0.1, i.e., less than one. Was a fraction of metal-free/apo-enzyme considered when modelling the crystal structure? Apo proteins may have some loops experiencing disorder (Kim et al, (2011), Plos One), an issue that is not discussed by the authors and may affect the quality of the structure.
4. Lines 323-324. The authors state "Additionally, we observed no bacteria in the dissected mice receiving combination therapy". It should be clarified that the observation was made by direct visual inspection of the infection site (Supplementary Fig. S9), and not by microscopic analysis of tissue samples from the infected mice.
5. Regarding the in-vivo infection studies, the general readership of the journal would benefit from a brief statement regarding the role of mucin in these experiments to aid in establishing intraperitoneal infections.
6. Additional experimental results are introduced within the Discussion (in lines 355-363), and should be moved to an appropriate location within the Results section.
7. As the authors themselves acknowledge, B3 enzymes (lacking a Cys ligand) may not be inhibited by Bi(III) compound. Mono-Zn(II) B2 enzymes have a Cys ligand, but their active site may be less accessible and rigid, and this fact may kinetically affect the Zn(II)-Bi(III) exchange. I thus suggest toning down a general application of this strategy to all MBLs and restrict the conclusions to B1 lactamases, which is already an outstanding achievement.

Minor comments:

- Lines 94 - 99: Use consistent verbal tenses ("were initially screened", "is confirmed to be", "is examined")
- Many instances of use of the incorrect phrase "for overnight" (e.g. lines 307, 572, 580)
- Line 90: Replace "beta-lactam antibiotic" with "beta lactam antibiotics"
- Line 96-67: Redaction
- Line 129: Replace "1000 folds" with "1000-fold"

- Line 130: Redaction
 - Line 164: "contributes primarily to Bi(III) ions"
 - Line 233: Bi-bound and Zn-bound of NDM
 - Line 254: Replace "Countered at 5.5" by "Contoured at 5.5"
 - Line 281: consider using "NDM-1-overexpressing E. coli Rosetta cells"
 - Line 283: replace "human" with "humans"
 - Line 341: "Thus alternative strategy" -> "Thus an alternative strategy"
 - Line 369: Explain what is meant by the phrase "rendering facial movement of these residues".
 - Lines 372-381: Improve redaction
- Figure 3: Labels for H189 and H120 in NDM structures are inverted

Reviewer #2 (Remarks to the Author):

The manuscript describes use of Bi(III) compounds to potentiate the action of carbapenem antibiotics against bacteria producing metallo-beta-lactamases (MBLs). MBLs confer resistance towards almost all beta-lactams and escape the action of beta-lactamase inhibitors in current clinical usage. The authors screened a panel of metallic compounds and identified Bi(III) as able to potentiate meropenem activity against bacteria producing the NDM-1 MBL, reducing the MIC of a clinical E. coli from 16 to 2 mg/l. Bi(III) binds to NDM-1, as evidenced by enzyme inhibition, ability to displace Zn(II) measured by ICP-MS, and X-ray crystallography; using ligands from each of the two Zn(II) sites and with a key contribution from the active site cysteine residue. Bi(III)-meropenem combinations were effective in reducing meropenem MICs of bacteria expressing other MBLs in vitro, potentiated meropenem action in in vitro and in vivo infection models and suppressed the emergence of resistance in passage experiments. The authors conclude that bismuth compounds, as already approved drugs for treating e.g. *Helicobacter pylori* infections, can find utility as treatments for infections by carbapenem-resistant strains of Enterobacteriaceae producing MBLs, in particular NDM-1.

This is a comprehensive study of the interaction of Bi(III) compounds with NDM-1 and producer bacteria and may find more rapid application in the clinic than comparable studies since Bi(III) compounds are already in clinical use. The manuscript is concise and for the most part clear. I have only a limited number of suggested alterations.

First, details of the bacterial strains used are lacking throughout. More information could be provided on the origins of the strains used, their resistance phenotypes and whether other beta-lactamases are present; and some nomenclature would help make clear which strains are used in which experiments. Examples of where clarification on this point is needed include page 4 line 95, page 5 lines 126-7 (what was the NDM-1 negative strain?) and the legends to Table 1 (these are clinical strains?) and Figure 4 (is the same E coli strain used in the mutation experiments, the in vitro and in vivo infection assays and is the same strain used in Figure 1)? Regarding the checkerboard assays shown in Figure 1 b and c, it would be useful to know if there is any relationship between the NDM-1 positive and negative strains. Ideally the comparison would be made with isogenic strains carrying either NDM-1 on a plasmid vector or a control plasmid with no insert. This point notwithstanding, the subsequent experiments do confirm the action of Bi(III) on NDM-1.

Second, the manuscript makes use of Fractional Inhibitory Concentration (FIC) determinations to describe potentiation of meropenem action by Bi(III). As described in Methods this requires determination of an MIC for the relevant Bi(III) compounds, but this was not evident. Did the authors determine MICs for these agents, or have FIC values been reached by assuming values for the Bi(III) MIC?

Third, there are no details in Methods describing how the values for K_d and K_d' were determined. The text makes reference to the Ryan-Weber equation (a citation is needed here) but this and subsequent calculations used to determine K_d' are not described.

Minor points:

Page 6 lines 143 – 145. In the absence of any data showing the effect of Bi(III) on meropenem action against MBL-producing non-fermenting species such as *Pseudomonas aeruginosa* or *Acinetobacter baumannii*, or that show activity against subclass B3 MBLs which lack the active-site Cys residue, it seems premature to conclude that “CBS is a potential broad-spectrum inhibitor of MBLs regardless of their (sub)types or the bacteria that host them” if Pa not included. I suggest qualifying this statement to make clear that this applies to B1 enzymes in Enterobacteriaceae.

It is not clear whether any of the Bi(III) compounds tested show antibacterial activity in the absence of meropenem. The data in Figures 1b and 1c suggest not, but Table 1 contains no information on any antibacterial activity of the Bi(III) compounds tested and there is no Bi(III) only control in Figure S1b.

Page 7 lines 179 -80 please provide a reference for the effect of the C208A mutation upon activity. This has been previously demonstrated for NDM-1 (PLoS ONE 6(8): e23606) and for other MBLs.

Pages 7 -8 lines 187 – 191: Did the authors carry out the cellular thermal shift assay for the C208A mutant? This would strengthen the assertion that the Cys residue is important to binding.

Page 8 lines 199 -201: In Figure 2e, was it possible to determine the metal content of the active form of NDM-1 present when the Bi(III) bound enzyme was treated with zinc? I.e what was the Zn(II) and Bi(III) content of this material?

Lines 237 – 238. It seems slightly misleading here to say that Bi(III) has a high propensity to bind in the Zn1 site. The legend to Figure 3d appears to provide a more accurate portrayal of the situation- i.e. “Bi(III) is located in between the two Zn(II) ions slightly closer to Zn1”.

Figure S5 legend: Please clarify which electron density map, at what contour level, is shown in the Figure.

Table S2: readability could be improved by better spacing out the strain information in this Table. What were the growth/induction conditions for NDM-1 expression in the BL21 and Rosetta strains, and what is the difference between the Rosetta (DE3) NDM-1+ and Rosetta(DE3)overexpressed NDM-1+ conditions (or strains)?

Page 11 Figure 4d. What was the concentration of meropenem used in the passage experiment? This information should be in the Figure legend, and was not clear in the Methods section.

Figure 4f: how many mice were used in each group?

Re: Manuscript number: NCOMMS-17-16035A-Z

“Bismuth antimicrobial drugs serve as broad-spectrum metallo- β -lactamase inhibitors”

We appreciate both reviewers' favorable comments and helpful suggestions!

Reviewer #1

Major comments:

(1) We truly appreciate the reviewer's comments and suggestions. Limited proteolysis of purified apo-, Zn-bound- or Bi-bound NDM-1 protein was performed as suggested. We confirm that the deprivation of Zn(II) leads to significant degradation in NDM-1, in consistence with what was stated in the mentioned work. Importantly we find that the displacement of Zn(II) with Bi(III) greatly induces the degradation of NDM-1 protein, which may partially contribute to suppression of the development of high-level resistance in NDM-1 producers by bismuth drug. We've added the data as Fig. S6 in revised version (Page 15, Supplementary Information). And related description is also incorporated into the main text.

(2) The cellular thermal shift assay only allows us to detect the soluble part of NDM-1 protein. Therefore, we clarify here that the melting temperature was measured for soluble part of NDM-1.

(3) Bi(III) was modelled into the anomalous electron density map and its occupancy was refined according to the anomalous signal. The combined occupancies of both Bi(III) conformations are around 0.6, meaning that there is indeed a fraction of metal-free enzyme in the structure. Because X-ray crystal structures are the average of all molecules in the crystals, the final refined structures include the conformational information of both Bi(III)-bound and metal-free enzyme molecules.

Although there is a fraction of metal-free enzymes in the structure, we did not observe obvious variations between the Bi(III)-bound enzyme and di-Zn(II) enzyme or metal-free enzymes reported previously (Kim et al, *Plos One* 2011). As stated in that paper, “most active loops show very small conformational changes with the exception of ASL1 and to some extent ASL4” for all nine structures including 5 metal-free and 4 mono-Zn(II) molecules. Our structures of Bi-bound and di-Zn(II) enzymes are consistent with this result and only subtle variations are observed in the ASL1 and ASL4 loops. We have included the comparison in the discussion part in the revised version (Page 13, main text).

(4) We thank this reviewer for helpful suggestion. We have revised the sentence and added “by direct visual inspection of the infection site” as suggested in the revised version (Page 22, Supplementary Information)).

(5) This reviewer's helpful comments are highly appreciated. A brief description of mucin's function was added into the main text of the manuscript accordingly in the revised version (Page 11, main text).

Mucin is the macromolecular component of mammalian mucus and was reported to enhance the bacterial pathogenicity when injected into the peritoneal cavity of animal decades ago (Olitzki L.

Bacteriol Rev, 1948, 12:149). The role of mucin on enhancing the bacterial virulence has not been fully unveiled. It could be attributed to (a) its coating effect on bacterial pathogens (b) its capability of lowering the levels of serum bactericidin or properdin (Dewitt CW, *J Bacteriol, 1958, 76:631*) (c) replenishment of nutrition for bacteria in mouse intraperitoneal infection model (Rodriguez CA et al., *Antimicrob Agents Chemother 2015, 59:233*). The pathogenicity of NDM-HK was much increased when 2% mucin was used, in which the lethal dose of NDM-HK was dropped from 5×10^7 CFU to 1×10^5 CFU per mouse. The mucin-assisted infection model used in this study was modified according to a report previously (Collins JJ, *Sci Transl Med 5, 2013, 190ra181*), whereas 8% mucin was used in their studies.

(6) We have moved these experimental results into Results section as suggested in the revised version.

(7) We agree with this reviewer that whether bismuth can displace Zn(II) from B2 enzymes and how fast such Zn(II)-Bi(III) exchange may warrant further study in future. We have revised our conclusion “We thus anticipate that CBS or other bismuth compounds can be repositioned or developed as a new class of broad-spectrum inhibitors for B1 of MBLs” as suggested in the revised version.

Minor comments.

(1) We thank this reviewer’s careful reading on the manuscript and have corrected these typo errors in the revised manuscript.

(2) We have made correction accordingly in the revised manuscript.

(3) We have corrected the mistakes in the revised manuscript as suggested.

(4) We have edited the sentence to make it easy to read as suggested in the revised manuscript

(5) We have made changes as suggested in the revised manuscript.

(6) We have revised the sentence as suggested.

(7) This has been changed to “is attributed primarily to Bi(III) ions” in the revised manuscript.

(8) We have corrected this to “Bi-bound and Zn-bound forms of NDM-1” in the revised manuscript.

(9) We have remade the figure and corrected this mistake in the revised manuscript.

(10) We agree with this referee and have made changes accordingly in the revised manuscript.

(11) The error has been corrected in the revised manuscript.

(12) We thank this reviewer’s careful reading and have made changes in the revised manuscript.

(13) We apologize for typo error on facial. It should be “facile movement of these residues”. We have reworded the related description to make it more readable.

(14) We have revised this part to make it easy to read in the revised manuscript.

(15) We have corrected the mistakes in the revised manuscript (Fig.3b-d and Fig. S8).

Reviewer #2

Major comments:

(1) We thank this reviewer's helpful comments and suggestions.

(i) The information of NDM-1 positive *E. coli*. The bacterial strain used in susceptibility test and time kill assay (Fig. 1b, d and f), resistance study, *in vitro* and *in vivo* infection assay (Fig. 4a, c, d, e and f) is the same one, which is denoted as the clinical isolate of NDM-1 positive *E. coli* in the original manuscript. This strain is the first NDM-1 producing *E. coli* isolate found in Hong Kong (Ho, PL et al., *Plos ONE* 6, 2011, e17989). The $bla_{\text{NDM-1}}$ gene is encoded in the plasmid, namely pNDM-HK. The complete sequence of pNDM-HK is shown below (Fig. 1 in aforementioned paper).

The $bla_{\text{NDM-1}}$ co-exists with other two β -lactamases genes, $bla_{\text{DHA-1}}$ and $bla_{\text{TEM-1}}$ in pNDM-HK. Both TEM-1 (class A) and DHA-1 (class C) are serine β -lactamase (SBL) and are susceptible to meropenem based on their mechanism. TEM-1 hydrolyzes penicillins and narrow spectrum of cephalosporins and was reported to have extremely low hydrolysis parameters towards meropenem (Queenan, et al *Antimicrob Agents Chemother*, 2010, 54:565). DHA-1 is AmpC β -lactamase and was also reported to be susceptible to meropenem. (Kim, JY et al., *Ann Clin Lab Sci*, 2004, 34: 214). Given this, we consider that the resistance to meropenem of this clinical isolate is endowed by $bla_{\text{NDM-1}}$ and the co-existence of other two SBLs would not lead to observable influence to the subsequent experiments.

For the sake of prudence, we inserted the gene of NDM-1 into pET-28a plasmid and transformed vector pET-28a-NDM-1 into either *E. coli* BL21(DE3) or *E. coli* Rosetta(DE3) to produce the NDM-1 producing strains with relative clean backgrounds. We performed the susceptibility test by using either meropenem, CBS or their combination and the results are given in the Table S3 in the revised manuscript. The MICs of MER against NDM-1 positive *E. coli* BL21(DE3) and *E. coli* Rosetta(DE3) were 16 and 32 $\mu\text{g mL}^{-1}$ respectively, which had 8-fold decrease for each strain to 2 and 4 $\mu\text{g mL}^{-1}$ when used in combination with CBS. This result is consistent with what we observed for the NDM-1 positive clinical isolate. Thus, in consideration of the pathogenicity and significance of the clinical isolate, we used this strain for the following cell-based and animal studies.

(ii) The NDM-1 negative strains

The NDM-1 negative strain is the plasmid-cured strain obtained by *in vitro* passage of NDM-1 positive clinical isolate *E. coli* in antibiotic free medium as described in the Bacteria section in the original methodology. The loss of NDM-1 was confirmed by PCR. It was also confirmed by susceptibility test with MER MIC of $0.03 \mu\text{g mL}^{-1}$, which is the same level as the susceptible reference strain, *E. coli* ATCC 25922. We have added the MIC of NDM-1 negative strain into the Table S3 in revised version. The NDM-1 negative strain is used in the susceptibility test (Fig.1c), resistance study (Fig. 4b and c) and the pre-experiment of animal study (Fig. S11).

As mentioned in the last part of this answer, we constructed two isogenic strains carrying either NDM-1 on a plasmid vector or a control plasmid with no insert. As shown in Table S3, the combined use of CBS led to no decrease in the MER MIC in the control strains (*E. coli* BL21(DE3) and *E. coli* Rosetta(DE3)), which is also consistent with what we observed in the NDM-1 negative strain made from clinical isolate.

(iii) To improve the readability, we use new nomenclatures to present all the bacterial strain used in the revised manuscript (Table S1) and the related paragraphs have been reworded accordingly.

(2) We set $256 \mu\text{g mL}^{-1}$ as the MIC of Bi(III) (except for $\text{Bi}(\text{NAC})_3$) for the determination of FIC values. For $\text{Bi}(\text{NAC})_3$, we set $128 \mu\text{g mL}^{-1}$ as MIC to determine the FIC against NDM-HK and set $256 \mu\text{g mL}^{-1}$ as MIC against other bacterial strains. The MIC values of all Bi(III) compounds for the NDM-1 positive *E. coli* (NDM-HK) have been incorporated in Table S2 in the revised manuscript.

(3) We have added detailed information into the method about fitting the data with Ryan-Weber equation to obtain K_d and the method to calculate K_d' from K_d as suggested in the revised manuscript and references are also added.

Minor comments:

(1) We agree with this reviewer's comments and have reworded the statement as "CBS is a potential broad-spectrum inhibitor of B1 MBLs regardless of their (sub)types or the Enterobacteriaceae that host them" in the revised manuscript.

(2) We truly thank the reviewer's help comments.

(i) The antimicrobial activity of Bi(III) alone has been tested and data are incorporated into the Table S2 in the revised version.

(ii) The purpose of the primary screen experiment is to rapidly screen out any compounds that would improve the antimicrobial activity of MER. In the pre-experiment we found that none of the metal compounds, including two active compounds, $\text{Bi}(\text{NIT})_3$ and gallium nitrate (Ga(II)) show observable antimicrobial activity against NDM-HK. In the following susceptibility assay, MICs of both $\text{Bi}(\text{NIT})_3$ and gallium nitrate are determined to be greater than $256 \mu\text{g mL}^{-1}$. We've included the related additional information in the legend of Fig. S1 in the revised manuscript.

(3) We apologized for missing out this reference, which has been added accordingly in the revised manuscript.

(4) We agree with this reviewer. We carried out this experiment but did not show in the original supporting information. Negligible changes from 48.24 °C to 49.72 °C on the melting temperature of NDM-1-Cys208A was noted when the bacterial cells carrying this mutant were treated with CBS, indicative of less target engagement of CBS. This result suggests the importance of Cys208 in the binding of bismuth to NDM-1 in cells. The data have been added as Fig. S4b and the related description has been made accordingly in the revised manuscript.

(5) This is very good suggestion. We have performed the experiment as suggested and the result is shown as Fig. S5 in the revised version. Addition of increasing concentration of Zn(II) to Bi-bound NDM-1 led to a slight decrease in the amounts of bismuth that bound to the protein, when two molar equivalents of Zn(II) relative to the protein was added, about 0.87 molar equivalents of Bi(III) still remains in the NDM-1 protein and *ca.* 0.17 molar equivalents Zn(II) was also observed to bind to NDM-1, suggesting that only small portion of Bi(III) can be replaced by Zn(II) from NDM-1.

(6) We have reworded the related description in the revised manuscript to make it easier to read.

(7) We have remade this figure contoured at 5.0σ to make the color consistent in the revised version.

(8) We thank this reviewer's helpful suggestion.

For the test of NDM-Rosetta OX, the growth condition briefly goes as follow. An overnight culture of NDM-Rosetta was 1000-fold diluted into fresh LB medium and regrew to $OD \sim 0.6$. The overexpression of NDM-1 was induced by adding 1 mM isopropyl β -D-1-thiogalactopyranoside (IPTG) for 4 hours at 37 °C. The following susceptibility test was performed as described in the Microdilution MIC and MBC assay section in the methodology by using the resulting bacterial pellets with the supplementation of 200 μ M IPTG. For the test of C208A-Rosetta, the growth condition briefly goes as follow. An overnight culture of C208A-Rosetta was 1000-fold diluted into fresh LB medium and regrew to $OD \sim 0.6$. The overexpression of NDM-1-C208A was induced by adding 200 μ M isopropyl β -D-1-thiogalactopyranoside (IPTG) for 18 hours at 25 °C. The following susceptibility test was performed as described in the Microdilution MIC and MBC assay section in the methodology by using the resulting bacterial pellets with the supplementation of 200 μ M IPTG.

The test conditions for other bacterial strains were exactly the same as described in the Microdilution MIC and MBC assay section in the methodology.

The description on growth/induction conditions for NDM-Rosetta OX and C208A-Rosetta has been added in related paragraphs in the revised manuscript.

(9) We agree with this reviewer that the concentration of meropenem is important for passage experiment.

Instead of using fixed concentration of meropenem, a modified multiple-step selection method was used to increase the occurrence of higher-resistant mutant during the resistance development. The method used in current manuscript is modified according to the following papers: (a) Ling, LL et al.

Nature, 2015, 517:455; (b) Bogdanovich T, 2005, 49(10):4210; (c) Drago, L., et al, *J Antimicrob Chemother*, 2005, 56:353.

Herein we take the selection experiment by MER as an example. Specifically, an overnight culture of NDM-HK was diluted to about 10^7 CFU mL⁻¹ in LB broth. One hundred μ L of the bacterial suspension was added into each well of the 96-well plate that was pre-supplemented with meropenem to make the final concentration of MER of 0.5-, 0.75-, 1-, 1.25-, 1.5-, 1.75-, 2-, 3-, 4-fold MIC respectively. The 96-well plates were incubated at 37 °C for 24 hours. Cultures from the second highest concentrations that allowed growth were performed 1:1000 dilution in fresh medium. One hundred μ L of the bacterial dilution was added into each well of a new 96-well plate that was pre-supplemented with same concentrations of MER (0.5-, 0.75-, 1-, 1.25-, 1.5-, 1.75-, 2-, 3-, 4-fold MIC) as previous one. Then the plate was incubated under identical condition. The serial passage was repeated for 20 passages. Notably, the concentrations of MER for selection were increased to 2-, 4-, 6-, 8-, 16-, 24-, 32-fold of MIC respectively after 12 passages because of the development of antibiotic resistance in bacteria.

One more thing should be clarified here, we used 16 μ g mL⁻¹ as 1-fold MIC for MER and set 2 μ g mL⁻¹ MER+32 μ g mL⁻¹ CBS as 1-fold MIC for the selection by MER and CBS combination to exert the same or similar selection pressure on bacteria.

The related description has been added in the legend of Fig. 4d and elaborated in more details in the revised version.

(10) We thank the reviewer's kind comments. In mucin-assisted infection model, for NDM-1 negative strain (NDM-HK PCV), 5 mice per group were used in vehicle control and monotherapy of MER; for NDM-1 positive strain (NDM-HK), 8 mice per group were used in vehicle control, monotherapy of MER and CBS, 12 mice per group were used in the combination therapy. In murine peritonitis model without mucin, 5 mice per group were used in vehicle control, monotherapy of MER or CBS, or combination therapy.

The related information has been added in the legend of Fig. 4f, S11 and S13 in the revised manuscript.

REVIEWERS' COMMENTS:

Reviewer #2 (Remarks to the Author):

The authors have made a comprehensive effort to address the reviewers comments. I have only a small number of suggestions arising:

The fact that MIC values for Bi(III) compounds were set to 256 ug/ml for the purposes of FIC determination should be included in Methods (p26)

In lines 213-216 the description of the proteolysis experiments would benefit from a little more explanation, which would also more directly address the comments of reviewer 1 regarding membrane localization of NDM-1.

The authors have added details of the contour level to the legend to Figure S8, but not the coefficients for the electron density map ($F_o - F_c$) shown. This information should be added.

We appreciate both reviewers' favorable comments and helpful suggestions!

Reviewer #2

(1) The fact that MIC values for Bi(III) compounds were set to 256 ug/ml for the purposes of FIC determination should be included in Methods (p26)

Answer: We thank this reviewer's helpful suggestions. We've added the description in the revised version (**Method**, Supplementary Information). The related description is also incorporated into the main text.

(2) In lines 213-216 the description of the proteolysis experiments would benefit from a little more explanation, which would also more directly address the comments of reviewer 1 regarding membrane localization of NDM-1.

Answer: This reviewer's helpful comments are highly appreciated. We have reworded the related description to make it more understandable.

(3) The authors have added details of the contour level to the legend to Figure S8, but not the coefficients for the electron density map ($F_o - F_c$?) shown. This information should be added.

Answer: We thank this reviewer's careful reading on the manuscript. The related information has been added as Fig. S8 and the related description has been made accordingly in the revised manuscript.